# Reducing the metabolic burden of rRNA synthesis promotes healthy longevity in *Caenorhabditis elegans*

Samim Sharifi [1,2,3,10], Prerana Chaudhari[2,10], Asya Martirosyan[2,7], Alexander Otto Eberhardt[1], Finja Witt[4], André Gollowitzer [4], Lisa Lange[1,2], Yvonne Woitzat[2], Eberechukwu Maryann Okoli[2], Huahui Li[2,8], Norman Rahnis [2], Joanna Kirkpatrick[2], Oliver Werz [5], Alessandro Ori [2,9], Andreas Koeberle [4], Holger Bierhoff [1,2,11] ✉ & Maria Ermolaeva [2,6,11] ✉

Ribosome biogenesis is initiated by RNA polymerase I (Pol I)-mediated synthesis of pre-ribosomal RNA (pre-rRNA). Pol I activity was previously linked to longevity, but the underlying mechanisms were not studied beyond effects on nucleolar structure and protein translation. Here we use multi-omics and functional tests to show that curtailment of Pol I activity remodels the lipidome and preserves mitochondrial function to promote longevity in *Caenorhabditis elegans*. Reduced pre-rRNA synthesis improves energy homeostasis and metabolic plasticity also in human primary cells. Conversely, the enhancement of pre-rRNA synthesis boosts growth and neuromuscular performance of young nematodes at the cost of accelerated metabolic decline, mitochondrial stress and premature aging. Moreover, restriction of Pol I activity extends lifespan more potently than direct repression of protein synthesis, and confers geroprotection even when initiated late in life, showcasing this intervention as an effective longevity and metabolic health treatment not limited by aging.

Aging is evolutionarily conserved, and four central nutrient-sensing pathways modulate healthspan and lifespan across taxa. The insulin/insulin-like growth factor signaling (IIS) and the mechanistic target of rapamycin (mTOR) promote anabolic reactions upon nutrient availability, whereas in a fasted state the adenosine monophosphate-activated protein kinase (AMPK) and the sirtuin family of nicotinamide adenine dinucleotide (NAD+)-dependent protein deacetylases trigger catabolic processes[1,2]. Shifting the balance from IIS and mTOR signaling towards AMPK and sirtuin activity by diverse interventions promotes longevity[1,2].

The IIS, mTOR, AMPK and sirtuin pathways impinge on Pol I-mediated transcription of rRNA genes (rDNA) into pre-rRNA, a precursor transcript comprising the three largest rRNAs[3]. Notably, Pol I activity accounts for the major part of cells' transcription and, together

[1]Institute of Biochemistry and Biophysics, Center for Molecular Biomedicine (CMB), Friedrich Schiller University Jena, Hans-Knöll-Str. 2, Jena 07745, Germany. [2]Leibniz Institute on Aging – Fritz Lipmann Institute, Beutenbergstrasse 11, 07745 Jena, Germany. [3]Matter Bio, Inc., Brooklyn, NY 11237, USA. [4]Michael Popp Institute and Center for Molecular Biosciences Innsbruck (CMBI), University of Innsbruck, 6020 Innsbruck, Austria. [5]Department of Pharmaceutical/Medicinal Chemistry, Institute of Pharmacy, Friedrich Schiller University Jena, Philosophenweg 14, 07743 Jena, Germany. [6]Cluster of Excellence Balance of the Microverse, Friedrich Schiller University Jena, Jena, Germany. [7]Present address: Cluster of Excellence Cellular Stress Responses in Aging-Associated Diseases, University of Cologne, Joseph-Stelzmann-Straße 26, 50931 Cologne, Germany. [8]Present address: Institute of Biomedical and Health Engineering, Shenzhen Institutes of Advanced Technology, 1068 Xueyuan Avenue, Shenzhen University Town, Shenzhen, PR China. [9]Present address: Genentech, 1 DNA Way, South San Francisco, CA 94080, USA. [10]These authors contributed equally: Samim Sharifi, Prerana Chaudhari. [11]These authors jointly supervised this work: Holger Bierhoff, Maria Ermolaeva. ✉e-mail: holger.bierhoff@uni-jena.de; maria.ermolaeva@leibniz-fli.de

with pre-rRNA processing and synthesis of ribosomal proteins, consumes a large portion of the cellular biosynthetic and energetic capacity[4,5]. Moreover, ribosome biogenesis is required for mRNA translation, placing pre-rRNA synthesis at the origin of the most energy-demanding cellular activities[4,6]. Two recent studies reported that perturbation of rRNA synthesis entails pro-longevity effects in *C. elegans* and *D. melanogaster*, either by inducing structural changes in the nucleolus, the organelle implicated in ribosome biogenesis, or by limiting protein synthesis, respectively[7,8]. However, the interplay between metabolic costs of Pol I activity and aging has not been explored in these studies. To link Pol I activity to cellular nutrient states, IIS, mTOR, and AMPK pathways target the essential Pol I transcription initiation factor 1A (TIF-1A, also known as TIF-IA or RRN3), which is conserved from yeast to human[9]. TIF-1A is inactivated by AMPK-mediated phosphorylation but requires IIS and mTOR-dependent phosphorylations to recruit Pol I to the rDNA promoter[10–12]. Thus, pre-rRNA synthesis and TIF-1A activity are likely to connect nutrient-sensing signaling and longevity regulation.

Dietary restriction (DR) slows down aging across model organisms from yeast to non-human primates by redirecting cellular energy utilization from biosynthesis towards maintenance and repair[2]. Notably, DR and DR-mimetic drugs like metformin were recently found to lose their geroprotective efficacy in late life due to aging-associated decline of metabolic plasticity[13,14]. Thus, novel pro-longevity compounds and interventions without age restriction are needed.

In this study, we identified an anti-correlation between rDNA transcription and longevity. Using genetics, omics, and functional assays in *Caenorhabditis elegans* and human primary fibroblasts, we showed that curtailment of pre-rRNA synthesis not only inhibits translation but also moderates energy expenditure and supports metabolic health during aging. The geroprotective efficacy of Pol I restriction does not decline with age, demonstrating its possible late-in-life use as a lifespan and healthspan extension treatment.

## Results

### Levels of pre-rRNA synthesis control lifespan and healthspan

To increase TIF-1A levels (encoded by the *tif-1A/C36E8.1* gene in *C. elegans*), we generated a strain harboring the *C36E8.1* cDNA fused to a Ty1-epitope tag (Supplementary Figs. 1a and 17a, b). Reverse transcription followed by quantitative PCR (RT-qPCR) revealed a ~2.5-fold increase in total *tif-1A* mRNA levels, in turn elevating pre-rRNA levels by ~1.5-fold (Fig. 1a, b). Notably, *tif-1A* overexpression conferred lifespan shortening, indicating that enhanced pre-rRNA synthesis accelerates aging in nematodes (Fig. 1c). To test if restraining rDNA activity has the opposite effect, we used RNA interference (RNAi)-mediated gene knockdown (KD) to reduce *tif-1A* expression. Compared with the non-targeting RNAi control, both *tif-1A* mRNA and pre-rRNA levels were efficiently diminished when worms were reared for two generations on double-stranded RNA (dsRNA) producing bacteria (Fig. 1d, e). In addition, we observed that the *tif-1A* KD reduced nucleolar size (Supplementary Fig. 1b, c), a trait that correlates with ribosomal biogenesis and is linked to longevity[7,8]. Accordingly, the lifespan of TIF-1A-depleted worms was extended by approximately 30% (Fig. 1f). To further confirm life extension through reduced Pol I activity, we used RNAi against *rpoa-2*, encoding the second largest Pol I subunit[15]. When initiated in L4 larvae, the *rpoa-2* KD was efficient after two days, eliciting a reduction in *rpoa-2* mRNA as well as pre-rRNA levels (Fig. 1g, h). In accord with the *tif-1A* KD phenotype, *rpoa-2* RNAi significantly extended lifespan (Fig. 1i), corroborating the geroprotective effect of restricted pre-rRNA synthesis in adult animals.

In *C. elegans*, the RNA-binding protein NCL-1 inhibits pre-rRNA processing and in turn ribosome biogenesis by downregulating the rRNA 2'-O-methyltransferase fibrillarin/FIB-1[16]. We found that NCL-1 also dampens pre-rRNA synthesis, RNA-mediated KD of *ncl-1* elevating

pre-rRNA levels and, in line with previous work[8], shortening lifespan (Fig. 2a–c). Thus, *ncl-1* RNAi had similar molecular and physiological effects as *tif-1A* overexpression, and both manipulations contrasted the effects of *tif-1A* and *rpoa-2* KDs.

Measurement of body size revealed that on the second day of adulthood (AD2, young adult animals), *ncl-1* KD worms were bigger than the control animals, whereas TIF-1A-depleted worms were significantly smaller (Fig. 2d). At old age (AD12), body size was increased and equalized across treatment groups (Fig. 2d), suggesting that *tif-1A* RNAi worms gained on the other two groups with age. We next carried out a neuromuscular performance (NMP) test monitoring the ability of worms to burrow through a pluronic gel towards bacteria applied on the top[17] (Fig. 2e). At AD2, the NMP correlated with the extent of rRNA gene transcription, i.e., NCL-1 and TIF-1A depletion conferred superior and inferior neuromuscular fitness, respectively (Fig. 2f). However, the fitness of TIF-1A-depleted nematodes remained stable throughout life while the performance of control animals, and even more so of the *ncl-1* RNAi group, declined, rendering *tif-1A* KD worms the fittest at AD12 (Fig. 2g). Given the observed fitness differences, we next assessed intestinal integrity as another health parameter associated with longevity[18]. We fed worms with Brilliant Blue, which only exits the intestine when its barrier function is compromised, causing whole body staining ('Smurf' phenotype)[19]. While barely detectable at young age, Smurf formation increased at AD12, being the highest upon NCL-1 depletion (~80%) and lowest in TIF-1A-deficient worms (~40%) (Fig. 2h, i). Notably, a previous study in *D. melanogaster* also connected reduced Pol I activity with improved intestinal integrity at old age[7], demonstrating the evolutionary conservation of this phenomenon. These findings show that elevation of pre-rRNA synthesis confers a fitness advantage early in life but accelerates health decline with age, while its restriction limits early life fitness transiently and eventually prolongs both lifespan and healthspan.

### rDNA activity modulates ribosome biogenesis and metabolism

To address the mechanisms causing inverse correlation between pre-rRNA levels and longevity, worms treated with control, *tif-1A,* and *ncl-1* RNAi were collected at AD2, AD6, and AD12 (young, middle and old age, respectively), and proteomes were profiled by mass-spectrometry detecting ∼5000 proteins in each sample (Supplementary Fig. 2). Principal component analysis (PCA) revealed differences between the age groups and RNAi treatments, with *tif-1A* KD animals clustering apart from controls at all ages, while *ncl-1* KD exhibited some overlap with controls (Supplementary Fig. 3). Notably, the two KDs shifted from controls in opposite directions, underscoring their different effects on lifespan. We next performed pairwise comparisons either of AD12 versus AD2 in the control RNAi group, or of *tif-1A* and *ncl-1* RNAi versus control RNAi at AD2, 6, and 12, thereby monitoring age- or rDNA activity-dependent proteome remodeling, respectively (Supplementary Data 1). In the initial targeted analysis, we focused on markers of adaptive stress responses previously linked to longevity, i.e., immune response (C-type lectins; CLECs), heat stress response (heat shock proteins; HSPs), and oxidative stress response (glutathione-S-transferases; GSTs)[14]. While the expression of several individual markers was significantly changed, we found that *tif-1A* RNAi treatment did not lead to an overall modulation of these three responses (Fig. 3a and Supplementary Data 2). Similarly, *ncl-1* KD worms showed no global changes in CLECs and HSPs but had persistently low expression of GSTs, indicating alterations in redox balance, which could impact mitochondrial homeostasis[20] (Fig. 3b and Supplementary Data 3). In contrast to the stress response pathways, ribosomal proteins and the lipid droplet-associated vitellogenins (VITs)[21,22] were strongly reduced by *tif-1A* KD (Fig. 3a and Supplementary Data 2), whereas in *ncl-1* RNAi exposed nematodes, the downregulation of VITs was less pronounced and ribosomal content became moderately upregulated with age (Fig. 3b and Supplementary Data 3).

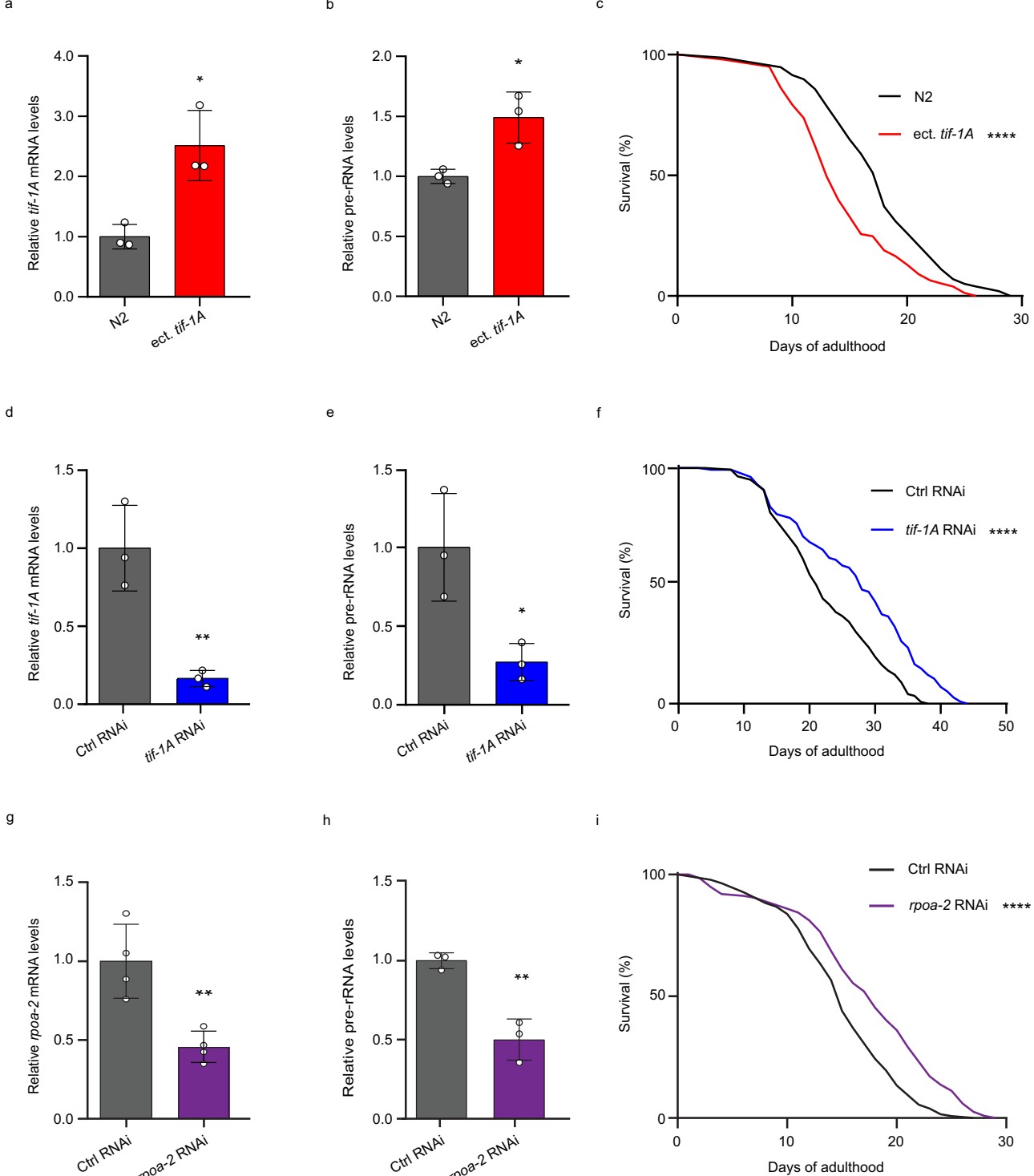

**Fig. 1 | Attenuation of Pol I activity increases lifespan. a, b** Comparison of *tif-1A* mRNA (**a**) and pre-rRNA (**b**) levels in *C. elegans* wild-type (N2 Bristol) strain and COP2239 strain harboring an ectopic *tif-1A* gene (ect. *tif-1A*). RT-qPCR data are shown relative to N2 and normalized to *tgb-1* (tubulin gamma chain) mRNA. Each data point represents an independent biological sample of *n* = 60 worms. **c** Lifespan analysis of N2 and COP2239 (ect. *tif-1A*) strains, *n* = 140 worms. **d, e** RT-qPCR analysis of *tif-1A* mRNA (**d**) and pre-rRNA (**e**) levels in nematodes treated either with control (Ctrl) RNAi or *tif-1A* RNAi. Data are relative to Ctrl RNAi and normalized to *tgb-1* mRNA. RNA from 3 independent experiments with *n* = 60 animals per sample was analyzed. **f** Lifespan analysis of animals treated with Ctrl RNAi or *tif-1A* RNAi, *n* = 140 worms. **g, h** Relative levels of *rpoa-2* mRNA (**g**) and pre-rRNA (**h**) in

nematodes exposed to control (Ctrl) or *rpoa-2* RNAi. Samples from 4 (**g**) or 3 (**h**) biological replicates (*n* = 60 animals per sample) were used. **i** Lifespan analysis of worms treated with Ctrl RNAi or *rpoa-2* RNAi, *n* = 140 worms. For **a, b, d, e, g** and **h** data represent mean values ± S.D. *P* values were determined using a Student's unpaired *t*-test. For lifespan analyses in **c, f** and **i**, results representative of 3 independent experiments are shown. Survival was scored daily. *P* values were calculated using the Mantel-Cox test. ****P < 0.0001, **P < 0.01 and *P < 0.05. All statistical tests used were two-sided, the exact *P* values and statistical analyses are reported in the Source Data file. *C. elegans* culturing temperatures and RNAi treatment durations for this and other figures are provided in Supplementary Data 23.

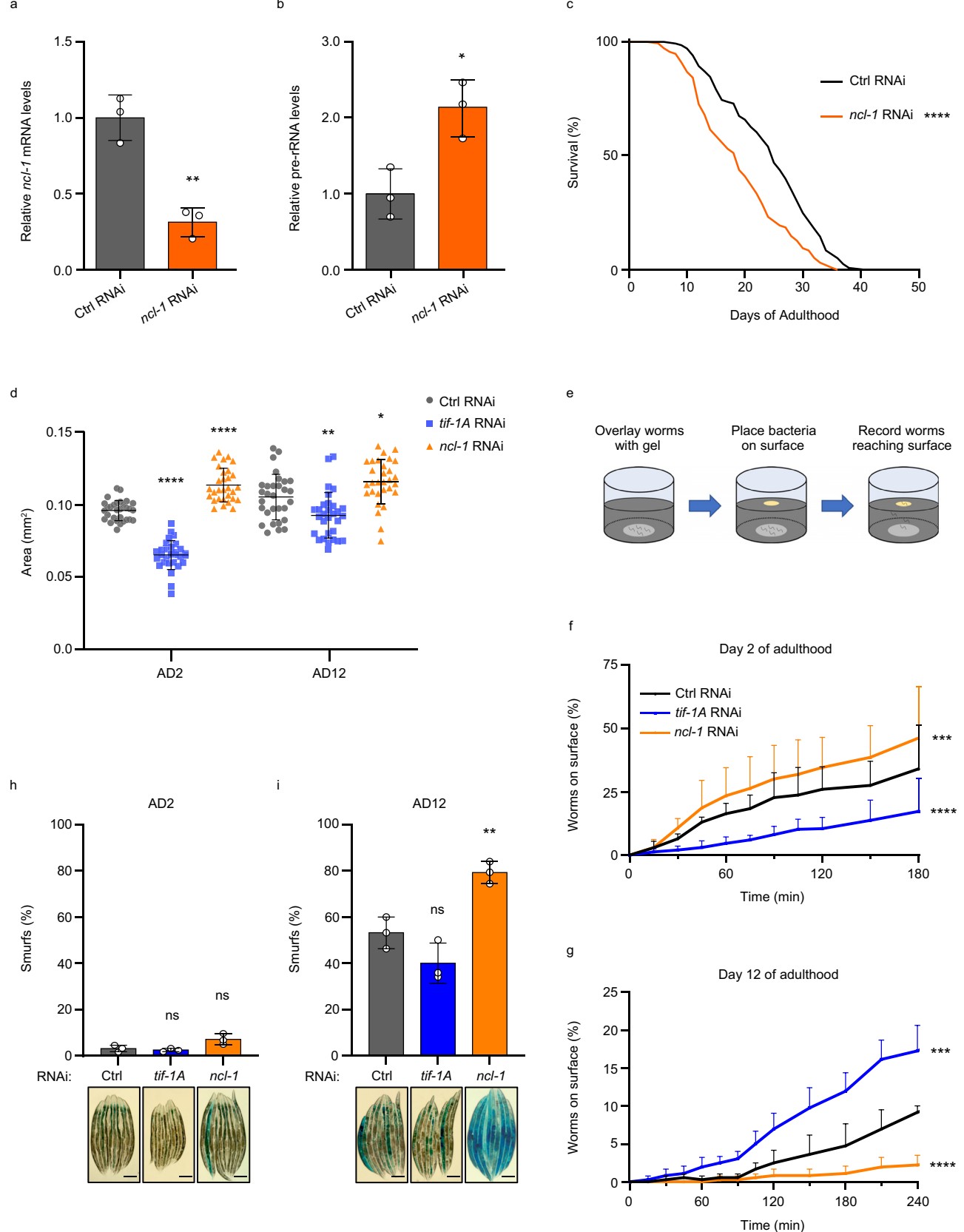

To assess the data by an additional unbiased method, the WormCat tool[23] was used to categorize proteins differentially expressed between old control and *tif-1A* RNAi or *ncl-1* RNAi worms (Supplementary Data 4–9). Ribosome and metabolism were indeed the two top enriched categories, (Supplementary Figs. 4 and 5; Supplementary Data 5 and 7), metabolism even surpassing ribosomal processes in the proportion of regulated proteins in both KDs (Fig. 3c, d; Supplementary Data 8 and 9). While changes in ribosomal biogenesis in animals harboring differential rDNA output were expected[7,8], the strong prevalence of changes in lipid and

**Fig. 2 | Pre-rRNA expression is inversely correlated with lifespan and health-span. a**, **b** RT-qPCR analysis of *ncl-1* mRNA (**a**) and pre-rRNA (**b**) levels of worms treated either with control (Ctrl) RNAi or *ncl-1* RNAi. Values are relative to Ctrl RNAi and normalized to *tgb-1* mRNA. Mean ± S.D. values of 3 independent experiments (*n* = 60 worms per sample) are shown. **c** Lifespan analysis of the nematodes upon treatment with Ctrl RNAi or *ncl-1* RNAi, *n* = 140 worms. Survival was scored daily and *P* values were calculated using the Mantel-Cox test. Result is representative of 3 independent trials. **d** Body size (area) measurement of animals treated with Ctrl, *tif-1A* or *ncl-1* RNAi. *n* = 30 worms. Data combine three independent trials and mean ± S.D. is shown. **e** Schematic of the locomotor performance test (burrowing assay). **f**, **g** Burrowing assays of animals treated with Ctrl, *tif-1A* or *ncl-1* RNAi. The % of animals reaching the surface was determined at the indicated time points. Data show mean values ± S.D. of 3 independent assays performed on the second day of adulthood (AD2) with *n* = 40–45 worms (**f**) or on AD12 with *n* = 40–60 worms (**g**). *P* values were calculated with the Mantel-Cox test. **h**, **i** Representative images (*n* = 10 worms per image) and quantification of intestinal integrity monitored by the Brilliant Blue dye retention assay. Worms with staining outside the gut ('Smurf' phenotype) were scored following treatment with Ctrl, *tif-1A* or *ncl-1* RNAi (*n* = 40–60 worms per treatment) at AD2 (**h**) or AD12 (**i**), scale bar 200 μm. Mean ± S.D. values of 3 independent experiments are shown. *P* values were determined using a Student's unpaired *t*-test. ns = not significant, ****$P < 0.0001$, ***$P < 0.001$, **$P < 0.01$, *$P < 0.05$. All statistical tests used were two-sided, the exact *P* values and statistical analyses are reported in the Source Data file.

mitochondrial metabolism was unexpected and warranted further investigation.

Previous studies have reported that the abundance of mitochondrial proteins, including components of the mitochondrial ribosome (mito-ribosome), decreases with age[24], which in turn functionally impacts longevity[14]. Accordingly, our analysis of control animals during aging confirmed these findings (Fig. 3e and Supplementary Data 10). Interestingly, mito-ribosomal content, which correlates with mitochondrial abundance, was differentially regulated by *tif-1A* and *ncl-1* RNAi, with *ncl-1* KD causing a strong increase in the abundance of mito-ribosomal proteins throughout life, and *tif-1A* KD promoting a moderate but significant elevation of mito-ribosomal protein levels with the biggest effect seen at old age (AD12) (Fig. 3f and Supplementary Data 11). This result suggested that rDNA activity impacts organismal fitness and longevity by modulating mitochondrial homeostasis.

## Curbed ribosome biogenesis delays metabolic aging

The persistently low abundance of ribosomal proteins in *tif-1A* RNAi animals (Fig. 3a) was paralleled by downregulation of total protein levels (Supplementary Fig. 6a). In line with the previously shown reduction of protein synthesis in long-lived flies with partially impaired Pol I function[7], this finding suggested that dampening of translation might contribute to life extension caused by *tif-1A* RNAi. In fact, reducing the translational output can counteract the loss of proteostasis, which is a hallmark of aging[25]. To elucidate to which extent *tif-1A* deficiency promotes longevity via translation inhibition, we combined *tif-1A* knockdown with the life-extending impairment of the *ife-2* gene encoding an isoform of the eukaryotic translation initiation factor 4E (eIF-4E)[26]. Lifespans of wild-type and *ife-2* mutant worms treated with either control or *tif-1A* RNAis, were assessed. Upon control treatment, the *ife-2* mutants lived moderately longer than wild-type animals, while *tif-1A* RNAi prolonged lifespan in both strains to similar levels (Fig. 4a). When we impeded translation elongation with cycloheximide (CHX), which also extends the lifespan of *C. elegans*[27], *tif-1A* RNAi was again more potent, increasing survival of worms similarly in the absence or presence of CHX (Supplementary Fig. 6b). These results indicate that the effects of *tif-1A* knockdown surpass those of translational inhibition by CHX or *ife-2* mutation with respect to lifespan extension. It cannot, however, be excluded that *tif-1A* KD reduces translation more potently.

Subsequently, we explored the connection between rDNA activity, protein synthesis, and energy metabolism, because rDNA transcription is not only a prerequisite for protein synthesis, but also initiates highly energy-demanding pre-rRNA processing and ribosome assembly[4,5]. Accordingly, we observed a ~1.5-fold elevation of adenosine triphosphate (ATP) content in TIF-1A-depleted worms relative to control animals (Fig. 4b). Notably, *ife-2* mutant worms and wild-type animals exhibited similar baseline ATP levels and showed comparably elevated ATP content upon *tif-1A* RNAi treatment (Fig. 4b), indicating that restriction of pre-rRNA synthesis may override translational

repression in terms of cellular energy conservation. The possible difference in the extent of translational inhibition between *ife-2* mutation and *tif-1A* RNAi treatment could also be contributing. Given the tight cross-regulation between synthesis of pre-rRNA and ribosomal proteins[28,29], we wondered whether depletion of the ribosomal protein RPS-15, which was previously reported to extend lifespan in *C. elegans*[30], elicits energy saving similar to *tif-1A* KD. In fact, *rps-15* RNAi treatment increased whole organism ATP levels, suggesting that perturbation of various steps of ribosome biogenesis reduces energy expenditure and prolongs lifespan (Supplementary Fig. 6c).

Aging-associated reduction of the cellular ATP pool has been observed in *C. elegans* and other animals contributing to the aging-associated metabolic decline[31]. Importantly, *tif-1A* KD was effective in counteracting this trait, and it boosted the ATP content in old KD worms (AD12) to levels comparable to young control animals (AD2) (Fig. 4c). It was recently shown that mitochondrial inhibitor metformin causes ATP exhaustion and toxicity in aged nematodes due to deterioration of their mitochondria[14]. As *tif-1A* depletion preserved both the mito-ribosome levels and the ATP content in late life (Figs. 3f and 4c), we tested whether it improved the metabolic resilience at old age by using metformin toxicity as a proxy readout[14]. In control animals, metformin administration from AD12 onwards induced the expected lethality and most animals died within 72 hours (Fig. 4d). In contrast, *tif-1A*-deficient worms survived much longer in the presence of metformin, although their median lifespan was also reduced relative to the untreated control (Fig. 4d). This result indicated that restriction of rRNA synthesis sustains energy metabolism and metabolic plasticity during aging in *C. elegans*.

Finally, we investigated if pharmacological inhibition of pre-rRNA synthesis can stabilize cellular energy metabolism similar to the genetic disruption of Pol I activity. We used metformin treatment to induce aging-like metabolic stress in human primary fibroblasts[14], and combined metformin exposure with Pol I inhibition by the compounds BMH-21 and CX-5461[32,33], with direct ATP repletion serving as a positive control. In accord with previous data[34], both inhibitors caused a dose-dependent decline of pre-rRNA levels (Supplementary Fig. 7a, b) as well as increased expression of the senescence marker p21 (Supplementary Fig. 7c, d). Consistent with the previously reported slowing of cellular growth and proliferation by Pol I inhibitors[34,35], BMH-21 and CX-5461 elicited a slight reduction of baseline mitochondrial activity as measured by the MTT assay, while strong ablation of mitochondrial function by metformin was alleviated by both drugs (Fig. 4e, f). These results demonstrated that stabilization of energy homeostasis and mitochondrial function by the reduction of rDNA activity is conserved between nematodes and humans, and can be achieved by pharmaceuticals. However, as both BMH-21 and CX-5461 induce cellular senescence, they are not suited for life-extension drugs. In addition, metformin treatment potentiates the cytotoxic effects of the CX-5461 in a murine lymphoma model[36]. This phenomenon is likely driven by specific alterations of cancer cell metabolism but it speaks against the possible co-administration of both drugs in the clinic.

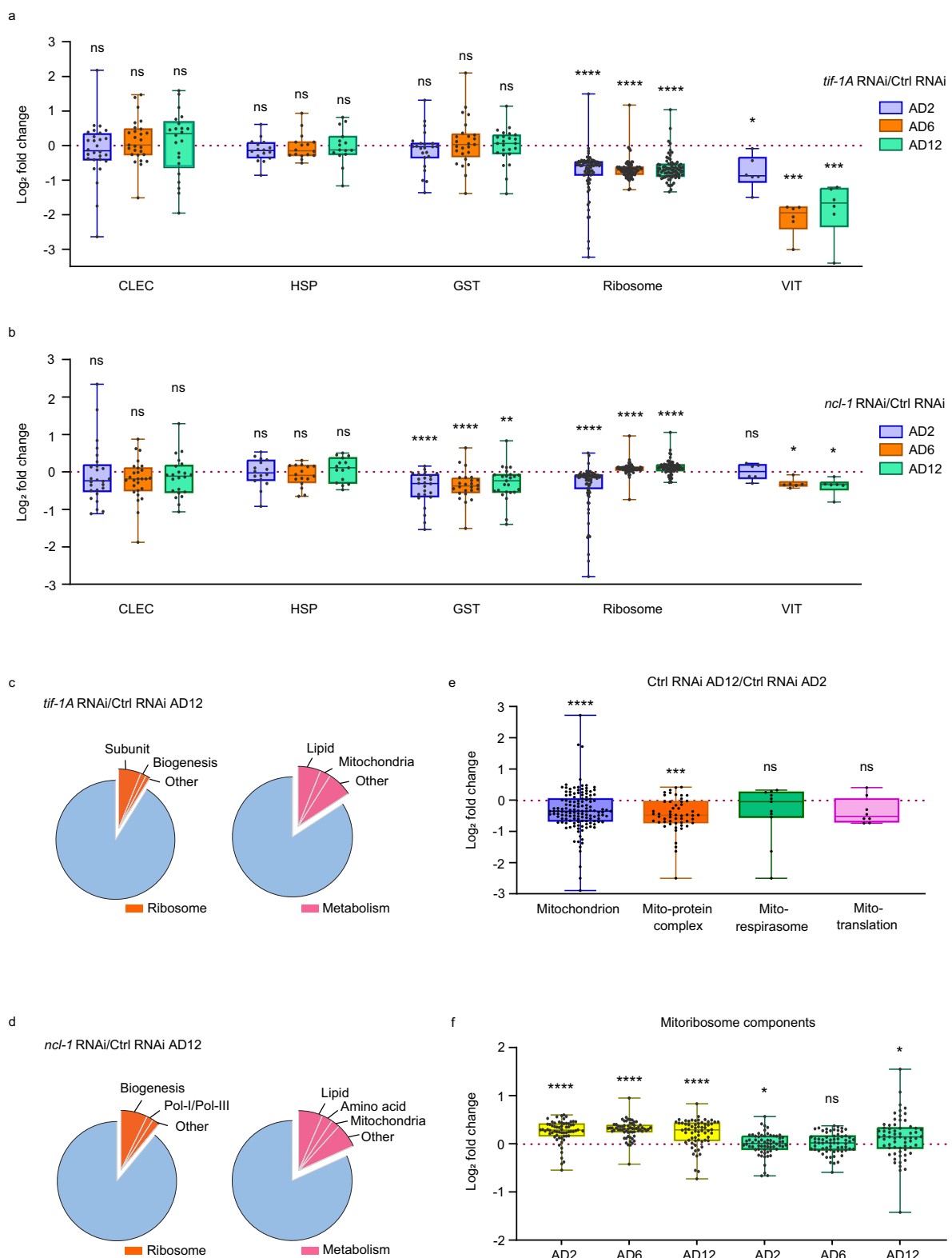

## rDNA activity contributes to lipidome changes in aging

Metformin toxicity has been linked not only to mitochondrial homeostasis but also to alterations in lipid metabolism[14,37]. As lipid metabolism was highlighted in the proteomics profiles of Pol I-modulated animals (Fig. 3c, d), we analyzed the lipidomes of control, *tif-1A* KD and *ncl-1* KD worms on AD2, AD6, and AD12. We found that in control animals fed with an HT115 *E. coli* diet,

the levels of phosphatidylcholines (PCs), phosphatidylethanolamines (PEs), and phosphatidylserines (PSs) increased between young and middle age and remained elevated also later in life (Fig. 5a and Supplementary Data 12). Interestingly, the aging-linked increase in phospholipids was abrogated in *tif-1A* KD worms, while *ncl-1* KD animals demonstrated their premature elevation on AD2, followed by an accelerated exhaustion on AD12

**Fig. 3 | Proteomics links Pol I activity, ribosomal content and metabolism.**
**a**–**f** Proteomes of nematodes treated with control (Ctrl) RNAi, *tif-1A* RNAi or *ncl-1* RNAi were analyzed at AD2, AD6 and AD12. Each condition has five biological replicates of *n* = 700 worms. **a**, **b** Box plots showing log2 fold changes of selected immune response (CLEC), heat stress response (HSP), oxidative stress response (GST), and ribosomal (Ribosome) proteins as well as vitellogenins (VIT), comparing *tif-1A* KD (**a**) and *ncl-1* KD (**b**) nematodes to age-matched controls. Each dot represents one protein, the median fold change within each group is shown as a bold line, the lower and upper limits of the boxplot indicate the first and third quartile, respectively, and the whiskers show the most up- and down-regulated proteins. The expression levels and Q values of individual proteins are reported in Supplementary Data 1–3. **c**, **d** Proteins differentially expressed (log2 fold change > 0.58, Q value < 0.05) in *tif-1A* KD (**c**) and *ncl-1* KD (**d**) versus Ctrl RNAi animals on AD12 were classified using WormCat, and the two top enriched

categories (Ribosome and Metabolism) with their subgroups are presented (analysis details are shown in Supplementary Data 8 and 9). **e** Box plots showing log2 fold changes of selected mitochondrial proteins between AD2 and AD12 Ctrl RNAi animals are presented. Either all measured mitochondrial proteins ('mitochondrion') or distinct categories ('mito-protein complex', 'mito-respirasome' and ('mito-translation') are shown. **f** Box plots showing log2 fold changes of selected mito-ribosome components in *tif-1A* RNAi and *ncl-1* RNAi worms compared to age-matched controls. The expression levels and Q values of individual proteins are reported in Supplementary Data 10 (for **e**) and 11 (for **f**). In **a**, **b**, **e** and **f** the Wilcoxon rank-sum test was used for statistical analysis; Bonferroni correction was performed to adjust for multiple comparisons. ns = not significant, ****P < 0.0001, ***P < 0.001, **P < 0.01, *P < 0.05. All statistical tests used were two-sided, the exact P values and statistical analyses are reported in the Source Data file.

(Fig. 5a), revealing differential lipidome trajectories between the two KDs.

We next examined the lipidomes of *tif-1A* and *ncl-1* KD nematodes for features that were previously linked to increased longevity, i.e., elevated levels of monounsaturated fatty acids (MUFAs) as well as reduced levels of oxidative damage-prone polyunsaturated FAs (PUFAs) and lipotoxicity-associated saturated FAs (SFAs)[38–40]. This pattern was indeed observed in long-lived *tif-1A* KD animals across age points and lipid classes, while *ncl-1* KD showed an opposite trend consistent with the shortened lifespan of these animals (Fig. 5b and Supplementary Fig. 8a; Supplementary Data 13–16). In addition, the analysis of the relative phospholipid proportions showed different trends between the two KDs: the percentages of PCs and PSs were elevated and the fractions of PEs and PGs were reduced in aged *tif-1A* KD worms, while opposite changes were seen upon *ncl-1* KD (Supplementary Fig. 8b–g and Supplementary Data 17). Notably, a higher PC/PE ratio has previously been linked to extended longevity as PCs are less prone to oxidative damage compared to PEs[38]. At the level of distinct lipid classes, an increase of the MUFA/PUFA ratio was observed in *tif-1A* KD animals, with particular significance seen for PCs and PEs (Fig. 5c, Supplementary Fig. 9a, b and Supplementary Data 18, 19). As MUFAs are less susceptible to peroxidation, a high MUFA/PUFA ratio entails pro-longevity effects[39,40]. Contrary to the effect of *tif-1A* RNAi, the *ncl-1* RNAi administration caused a premature decline of MUFA content across lipid types (Fig. 5c, Supplementary Fig. 9a, b and Supplementary Data 18, 19). In addition, the relative levels of SFAs in distinct lipid types showed trends of elevation in *ncl-1* KD animals and this trend was inverted in *tif-1A* inactivated worms (Supplementary Fig. 9c and Supplementary Data 18), again consistent with opposite effects on longevity of the two KDs. Finally, we assessed specific triacylglyceride (TAG) features that were previously identified as markers of metabolic aging and mitochondrial dysfunction, i.e., increased FA chain length and number of double bonds higher than three (*n* ≥ 3 PUFAs)[40]. Importantly, both features were reduced by *tif-1A* RNAi treatment, and they were prematurely increased in *ncl-1* KD animals (Supplementary Fig. 10a, b and Supplementary Data 19).

Collectively, the lipidomics profile of *tif-1A* KD was consistent with attenuated aging while *ncl-1* KD animals showed elevated levels of lipids that are prone to oxidative damage (such as PUFA-containing PEs) and enrichment in signatures previously associated with aging-induced mitochondrial dysfunction.

**Pol I transcription promotes mitochondrial stress**

To test the link between rDNA transcription and mitochondrial stress, we examined GFP expression in young animals harboring a reporter (*hsp-6p::GFP*) for the stress-induced mitochondrial chaperone gene *hsp-6*[41] exposed to non-targeting, *tif-1A*, or *ncl-1* RNAi. We found that increasing pre-rRNA synthesis by *ncl-1* RNAi exacerbated mitochondrial stress, while stress was alleviated by the Pol I-constraining *tif-1A* RNAi (Fig. 5d and Supplementary Fig. 11). Mitochondrial stress in

young *ncl-1* KD animals was corroborated by elevated proportions of fragmented and very fragmented mitochondria (Supplementary Fig. 12a, b)—a phenotype associated with metabolic aging[14,42]. Interestingly, we observed a modest elevation of mitochondrial fission upon reduction of Pol I activity (Supplementary Fig. 12a, b), prompting us to investigate the link between remodeling of mitochondria, rRNA synthesis, and longevity. Nematodes deficient in either mitochondrial fusion (*fzo-1* mutant) or fission (*drp-1* mutant)[43] were not hampered in lifespan extension upon exposure to *rpoa-2* RNAi, demonstrating that mitochondrial remodeling is not relevant for the crosstalk between rDNA and lifespan (Supplementary Fig. 12c, d).

To test if perturbation of Pol I triggers mitochondrial stress responses, we knocked down *rpoa-2* in worm strains lacking key regulators of either the mitochondrial unfolded protein response (UPR^mt) (*atfs-1*)[44] or mitophagy (*pink-1*)[45], but observed the same relative lifespan extension by *rpoa-2* RNAi in wild-type and mutant animals (Fig. 5e and Supplementary Fig. 13a). Concurrently, our lipidomic profiling revealed enrichment of the stress-limiting lipokine PI(18:1/18:1)[46] by *tif-1A* KD, especially in old worms, while it was depleted by *ncl-1* KD throughout life (Supplementary Fig. 13b and Supplementary Data 20). Although a direct influence of PI(18:1/18:1) on mitochondria has not been shown, its pleiotropic effect on several cellular stress responses[46] may contribute to differential mitochondrial health of the KD animals.

To further test the role of mitochondrial integrity in life extension by lowered rDNA activity, we combined *rpoa-2* RNAi with the dose-dependent poisoning of mitochondria by the uncoupling agent FCCP[47]. Interestingly, the full longevity benefit of Pol I perturbation was maintained only at the lowest concentration of the drug, while higher FCCP doses diminished and even reversed the life extension of *rpoa-2* KD animals (Fig. 5f and Supplementary Fig. 13c, d). These findings confirm the key role of mitochondria in life extension by reduced Pol I activity and corroborate the ability of Pol I repression to buffer moderate mitochondrial stress, without involvement of UPR^mt and mitophagy.

Finally, we tested if longevity extension by reduced rDNA transcription engages a conserved transcription factor SKN-1/Nrf2 with known roles in protective mechanisms such as mitochondrial biogenesis[45], oxidative stress response[48], and lipid metabolism[38]. By combining *rpoa-2* RNAi with a loss-of-function mutation of *skn-1*, we found that Pol I restriction shortened the lifespan of *skn-1* mutants contrary to the prolongation seen in wild-type controls (Fig. 5g), demonstrating a functional link between rDNA transcription and SKN-1-mediated cellular responses including modulation of mitochondrial health.

**TAG turnover connects Pol I activity to mitochondrial stress**

We next explored the hierarchical relationship between mitochondria and lipid metabolism in mediating longevity benefits of reduced Pol I activity. As a starting point, we revisited the lipidome profiles of control, *tif-1A* KD, and *ncl-1* KD animals at different ages (Fig. 5a).

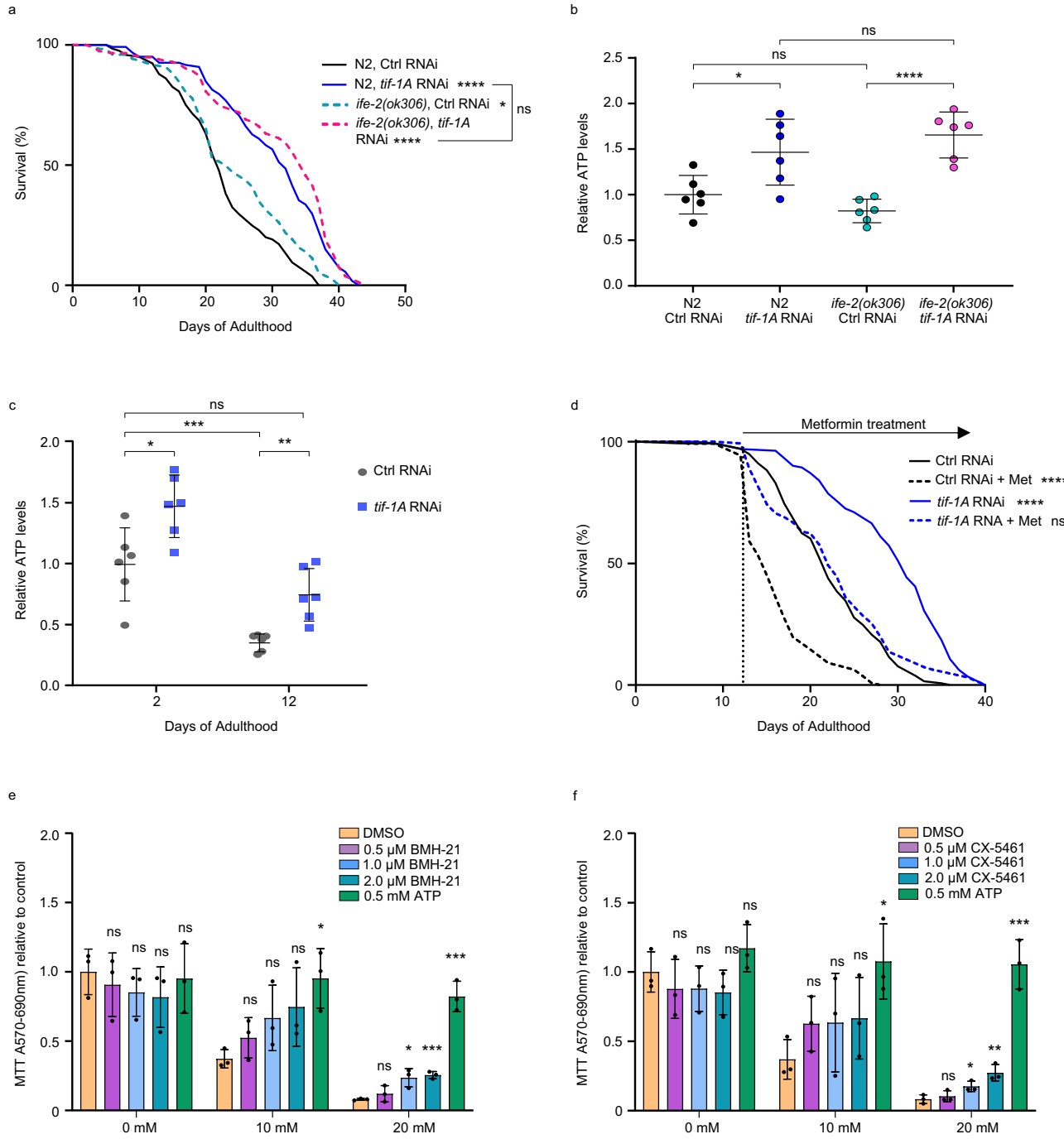

**Fig. 4 | Pol I inhibition decelerates metabolic aging. a** Survival analysis of *C. elegans* wild-type (N2 Bristol) and *ife-2* mutant strains treated with control (Ctrl) RNAi or *tif-1A* RNAi, *n* = 140 worms. **b** Comparison of ATP pools at AD2 between wild-type (N2) and *ife-2* mutant worms treated with Ctrl RNAi or *tif-1A* RNAi. ATP levels were measured in a bioluminescence assay and normalized to total protein content. Values are shown relative to N2 Ctrl RNAi. Data are from 6 independent experiments with *n* = 50 worms in each sample. Mean ± S.D. values are shown. **c** Comparison of ATP pools at AD2 and AD12 between worms treated with Ctrl RNAi or *tif-1A* RNAi. ATP measurement was conducted as in **b**, and values are relative to Ctrl RNAi on AD12. Mean ± S.D. values of 6 independent experiments with *n* = 50 worms are shown. **d** Survival analysis of worms exposed to Ctrl RNAi or *tif-1A* RNAi and, from AD12 onwards, to 50 mM metformin (Met), *n* = 140 animals. A

representative result of four independent experiments is shown. For **a** and **d**, survival was scored daily and *P* values were determined using the Mantel-Cox test. For **b** and **c**, a Student's unpaired *t*-test was used for statistics. **e**, **f** BJ primary human skin fibroblasts were treated with indicated doses of metformin in the absence (DMSO) or presence of the indicated concentrations of ATP (positive survival control) or the Pol I inhibitors BMH-21 (**e**) and CX-5461 (**f**). Mitochondrial activity was assessed by MTT assay, and an unpaired Student's *t*-test was used to compute *P* values, comparing each co-treatment to the respective DMSO + Metformin control condition, data represent mean ± S.D of 3 independent experiments. ns = not significant, ****$P < 0.0001$, ***$P < 0.001$, **$P < 0.01$, *$P < 0.05$. All statistical tests used were two-sided, the exact *P* values and statistical analyses are reported in the Source Data file.

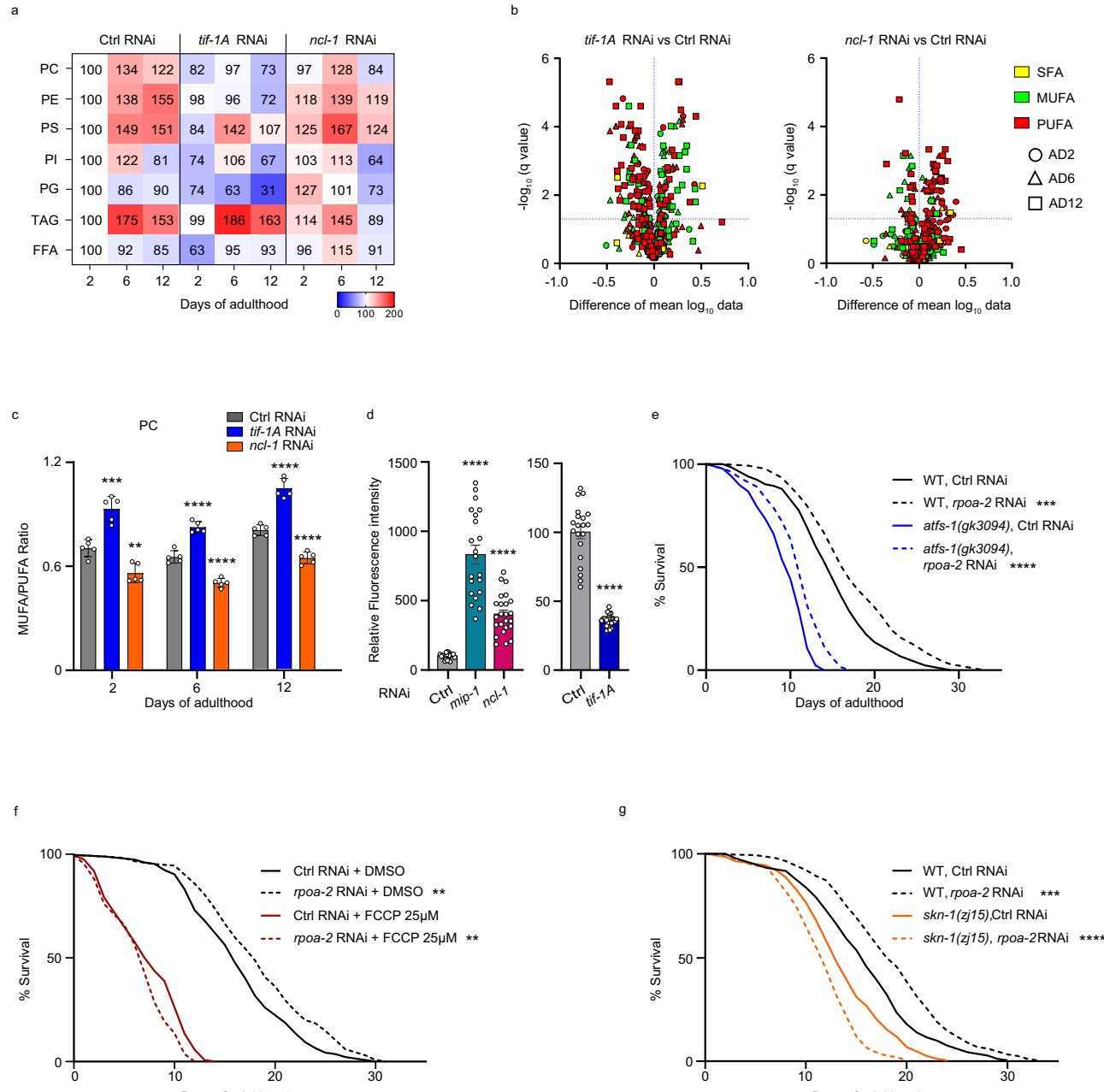

**Fig. 5 | Pol I activity affects the lipidome and mitochondria. a** Lipid abundance in control (Ctrl) RNAi, *tif-1A* RNAi and *ncl-1* RNAi animals at different ages, normalized to the internal standard and animal number (*n* = 700). Values were computed from absolute intensities; mean of 5 biological replicates expressed as % of Ctrl RNAi treatment at AD2 is shown in each case (see Supplementary Data 12 for raw data). PC phosphatidylcholine, PE phosphatidylethanolamine, PS phosphatidylserine, PI phosphatidylinositol, PG phosphatidylglycerol, TAG triacylglycerol, FFA free fatty acid. **b** Volcano plots of lipid species separated by age (AD 2, 6, 12) and fatty acid desaturation. SFA saturated fatty acid, MUFA monounsaturated fatty acid, PUFA polyunsaturated fatty acid. Horizontal axis subtracts Ctrl RNAi mean log₁₀ values from respective *tif-1A* RNAi (left panel) or *ncl-1* RNAi (right panel) values for each lipid, with statistics performed by multiple unpaired *t*-test on log-transformed mean values using a two-stage linear step-up procedure by Benjamini, Krieger and Yekutieli, FDR = 5%, data are from 5 biological replicates with *n* = 700 worms each. Calculations (from relative intensities) are shown in Supplementary Data 13–15.

**c** Ratio of MUFAs to PUFAs in PCs at indicated age upon treatment with Ctrl RNAi, *tif-1A* RNAi or *ncl-1* RNAi. Mean ± S.D. values from 5 replicates each with *n* = 700 worms are shown. Calculations (from relative intensities) are shown in Supplementary Data 19. **d** Fluorescence measurements in nematodes expressing *hsp-6p::GFP* following exposure to Ctrl, *mip-1*, *ncl-1* or *tif-1A* RNAi. *mip-1* RNAi served as a positive mito-stress control. *n* = 20-25 worms per condition. Representative result of 3 independent trials. For **c** and **d**, Student's unpaired *t*-test compared *ncl-1* and *tif-1A* RNAi effects to the age-matched Ctrl RNAi baseline. **e**–**g** Survival analysis of wild-type (N2 Bristol strain) and *atfs-1 (gk3094)* mutant nematodes (**e**), or N2 worms treated from AD1 onwards with the vehicle DMSO or 25 μM FCCP (**f**), or N2 and *skn-1 (zj15)* mutant nematodes (**g**), exposed to Ctrl or *rpoa-2* RNAi, survival was scored daily. Representative results of three independent experiments, *n* = 140 worms, statistics assessed by Mantel-Cox test. ns = not significant, ****P < 0.0001, ***P < 0.001, **P < 0.01, *P < 0.05. All statistical tests used were two-sided, the exact P values and statistical analyses are reported in the Source Data file.

Interestingly, we noticed that in the background of broadly reduced lipid content, the storage TAG levels were specifically elevated in *tif-1A* KD nematodes at old and middle age. Likewise, Oil Red O staining indicative of TAG levels[49] was increased in middle-aged *rpoa-2* KD animals (Supplementary Fig. 14a). In contrast, old and middle-aged *ncl-1* KD animals showed accelerated exhaustion of TAGs (Fig. 5a), indicating premature loss of the energy storage and revealing opposite regulation of TAG turnover between the two KDs, in line with their inverse longevity phenotypes. Importantly, the decline of TAGs, which reside in lipid droplets[22], was accompanied by the moderate reduction of the protein lipid droplet components VITs in *ncl-1* KD worms (Fig. 3b). Conversely, the strong downregulation of VITs in *tif-1A* RNAi animals (Fig. 3a) occurred amid stably high levels of TAGs, suggesting lipid droplet remodeling towards a higher lipid-to-protein ratio, which was previously linked to DR in facilitating effective regulation of TAG metabolism[50]. To test if TAG turnover is implicated in the differential longevity upon reduced or forced Pol I activity, we combined *rpoa-2* RNAi or *ncl-1* RNAi with knockdown of the *atgl-1* gene, encoding the *C. elegans* orthologue of the adipose triglyceride lipase (ATGL). ATGL-1 is implicated in mobilization of TAGs[51] and regulates responses to differential nutrient availability during dauer diapause and fasting[52,53]. Strikingly, both life extension of *rpoa-2* KD and lifespan reduction of *ncl-1* KD were reversed by co-inactivation of *atgl-1* (Fig. 6a, b), revealing the involvement of TAG metabolism in the longevity effects of differential Pol I activity. At the same time, dampening of peroxisome activity by mutation of *prx-5*, which encodes a peroxisome assembly factor[37,54] had no impact on life extension by lowered pre-rRNA synthesis (Fig. 6c). Moreover, interference with mitochondrial fatty acid β-oxidation by RNAi against *acs-2*[55], and curbing of fatty acid desaturation by KD of the Δ9 desaturase *fat-5*[39,56] did not affect the longevity benefits of reduced Pol I activity (Fig. 6d and Supplementary Fig. 14b–d). This neutral behavior of both interventions occurred despite a clear upregulation of ACS-2 and FAT-5 proteins in old *tif-1A* KD animals (Supplementary Fig. 14e and Supplementary Data 21), and the fatty acid profiles of *tif-1A* KD being consistent with elevated activity of Δ9 desaturases[38,39] (Fig. 5b and Supplementary Fig. 8a). These findings suggest that at least some of the lipidome remodeling events occurring in Pol I modulated animals are not causally linked to longevity, reemphasizing the specific role of TAG metabolism in Pol I-linked geroprotection.

We next tested if inhibition of *atgl-1* affects mitochondrial stress in *tif-1A* and *ncl-1* KDs. Interestingly, inactivation of *atgl-1* reversed high levels of *hsp-6p::GFP* expression in *ncl-1* RNAi-treated worms, while very low mito-stress levels of *tif-1A* KD nematodes were not further reduced by co-treatment with *atgl-1* RNAi, suggesting that ATGL-1 activity might be already lowered in response to Pol I repression (Supplementary Fig. 15a, b). These data reveal ATGL-1-dependent TAG lipolysis, which is likely elevated in *ncl-1* KD and reduced in *tif-1A* KD animals, to be an instigator of mitochondrial stress. Notably, increased lipolysis of TAGs and the subsequently increased flux of fatty acids into mitochondria were recently linked to alterations in mito-homeostasis and elevated oxygen consumption rate (OCR)[57]. We next measured OCR levels in *ncl-1* and *tif-1A* KDs at the age with the strongest difference in their TAG expenditure (AD12, Fig. 5a), and found oxygen consumption to be elevated in *ncl-1* RNAi-treated animals, while *tif-1A* KD worms showed diminished OCR levels (Fig. 6e). As OCR is a contributor to reactive oxygen species (ROS) production and oxidative stress[58], enhanced TAG lipolysis likely fuels the mitochondrial damage and the accelerated aging phenotype of NCL-1-depleted worms. Conversely, *tif-1A* KD animals exhibited reduced TAG turnover and lower OCR, presumably resulting in preservation of mitochondrial health and aging deceleration (Fig. 6f and Supplementary Fig. 16). However, as *atgl-1* RNAi treatment abrogates life extension of *tif-1A* KD animals (Fig. 6a), a basal level of TAG mobilization seems to be essential for health maintenance of *tif-1A*-depleted animals. This notion is supported by modestly increased mitochondrial stress levels of worms co-treated with *tif-1A* and *atgl-1* RNAis (Supplementary Fig. 15a, b). Finally, we found that depletion of SKN-1/Nrf2 not only abrogates longevity benefits of Pol I inhibition (Fig. 5g), but also limits the accumulation of TAGs induced by *rpoa-2* KD (Supplementary Fig. 14a). This observation suggests that SKN-1 might be implicated in the regulation of TAG turnover downstream of rDNA transcription, which is consistent with its previously reported role in lipid catabolism[59]. However, future studies are required to systematically validate this hypothesis.

## Pol I-linked geroprotection is not restricted by age

In light of recently uncovered age limitations of geroprotective interventions such as metformin and DR[13,14], we used the rapidity of the *rpoa-2* knockdown to explore at which stage in life attenuation of pre-rRNA synthesis is beneficial. First, we initiated the *rpoa-2* RNAi treatment at AD0, AD6, or AD8, and measured *rpoa-2* mRNA and pre-rRNA levels at AD10 (Fig. 7a–c). Both parameters were affected stronger by RNAi administered earlier in life, presumably reflecting the longer exposure to the dsRNA-producing bacteria and the higher rate of their ingestion early in adulthood[35] (Fig. 7b, c). Based on these results, we assumed that the timeline of survival experiments (~30 days) was sufficient for gene inactivation in all cases. Accordingly, the median lifespan was prolonged to a similar extent regardless of the stage in life when the *rpoa-2* knockdown was started (Fig. 7d), showing that initiating Pol I perturbation post-middle age is a robust pro-longevity intervention.

In nematodes, aging-associated tissue deterioration sets in shortly after the end of reproduction around AD6[60]. To test whether RPOA-2 depletion starting at AD6 or later can improve healthspan, burrowing assays were performed at AD10 after nematodes were transferred to *rpoa-2* RNAi at AD0, AD6, and AD8. Control worms were clearly outperformed by all three *rpoa-2* RNAi-treated groups (Fig. 7e). Similarly, Smurf assays showed that late-life RPOA-2 depletion improved intestinal integrity (Fig. 7f). Thus, although lowering of Pol I activity throughout adulthood was most beneficial, its initiation late in life still efficiently decelerated tissue deterioration and improved health.

## Discussion

Here, we uncovered that curtailment of pre-rRNA synthesis triggers multiple geroprotective adaptations in *C. elegans*, while elevation of rDNA activity confers an early advantage in fitness at the expense of premature aging. While previous reports of lifespan extension through reduced pre-rRNA and 5S rRNA synthesis focused on links to nucleolar size, nucleolar architecture and protein synthesis[7,8,61], we have discovered a novel metabolic remodeling response that mediates pro-longevity effects of Pol I impairment. Our in-depth analyses demonstrated that curtailment of pre-rRNA synthesis attenuates metabolic aging and stabilizes energy turnover, providing an ATP surplus for healthy homeostasis. Importantly, we found a similar energy preservation effect upon depletion of the ribosomal protein RPS-15 previously linked to life span extension in *C. elegans*[30]. Notably, no energy saving was detected upon restriction of translation initiation or elongation. While we cannot rule out that the curtailment of protein synthesis is more effective when targeting ribosome biogenesis instead of translation, the former will additionally reduce cellular ATP consumption by blocking ribosome assembly and restricting pre-rRNA synthesis and processing[4,5,29]. In line with this notion, distinct longevity mechanisms triggered by ribosomal protein depletion and inhibition of translation have been reported[30]. We also found that immune, proteostasis, and oxidative stress responses with known dependency on DAF-16/FOXO activity[62,63] were not changed in *tif-1A* and *ncl-1* inactivated animals. This finding corroborates observations of *daf-16* being dispensable for the longevity extension by the depletion of ribosomal proteins, while it is required for the survival effects of translational repression[30]. Specifically, our data demonstrate that

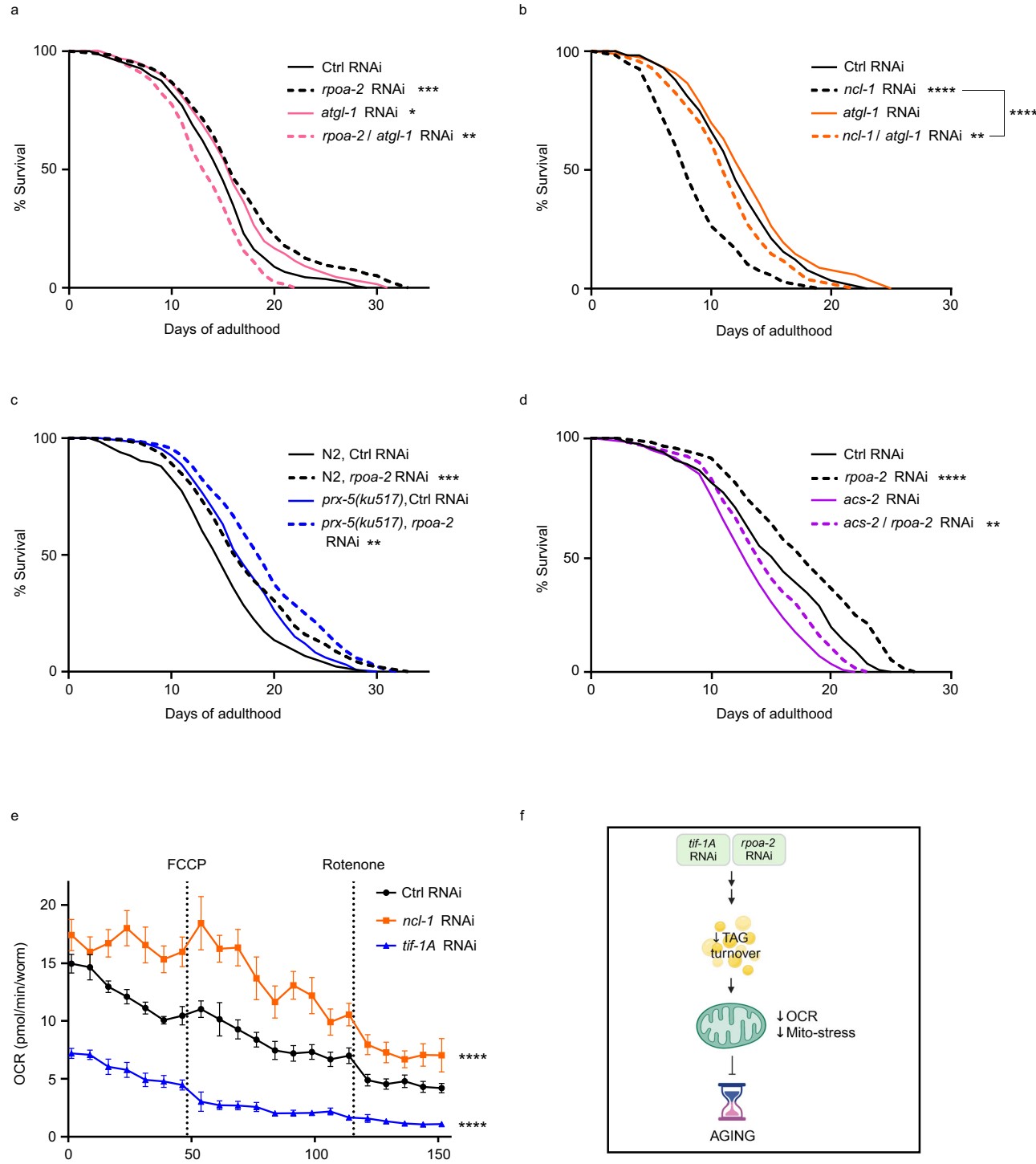

**Fig. 6 | TAG lipolysis mediates longevity effects of the differential Pol I activity.** **a**, **b** Survival analysis of wild-type (N2 Bristol strain) nematodes exposed to control (Ctrl), *rpoa-2* and *atgl-1* RNAi, or simultaneously to *rpoa-2* and *atgl-1* RNAi (**a**) or Ctrl, *ncl-1* and *atgl-1* RNAi, or simultaneously to *ncl-1* and *atgl-1* RNAi (**b**). **c** Survival analysis of N2 and *prx-5 (ku517)* mutant nematodes exposed to Ctrl or *rpoa-2* RNAi. **d** Survival analysis of wild-type worms exposed to Ctrl, *rpoa-2*, *acs-2* RNAi or double RNAi targeting *rpoa-2* and *acs-2*. Survival was scored daily, and *n* = 140 worms in all tests. Statistics were calculated using Mantel-Cox test. **e** Mitochondrial oxygen consumption rate (OCR) measured on AD12 in wild-type nematodes treated with Ctrl, *tif-1A* or *ncl-1* RNAi. OCR was normalized to the number of worms and mean ± S.E.M. of *n* = 85 animals is shown. Statistical analysis was performed by using unpaired *t*-test on area under the curve values. ns = not significant, ****$P$ < 0.0001, ***$P$ < 0.001, **$P$ < 0.01, *$P$ < 0.05. Representative results of 3 independent experiments are shown in all cases. All statistical tests used were two-sided, the exact $P$ values and statistical analyses are reported in the Source Data file. **f** Schematic representation of the molecular mechanism connecting curbed Pol I activity with the attenuation of metabolic aging. The image was created with BioRender.com.

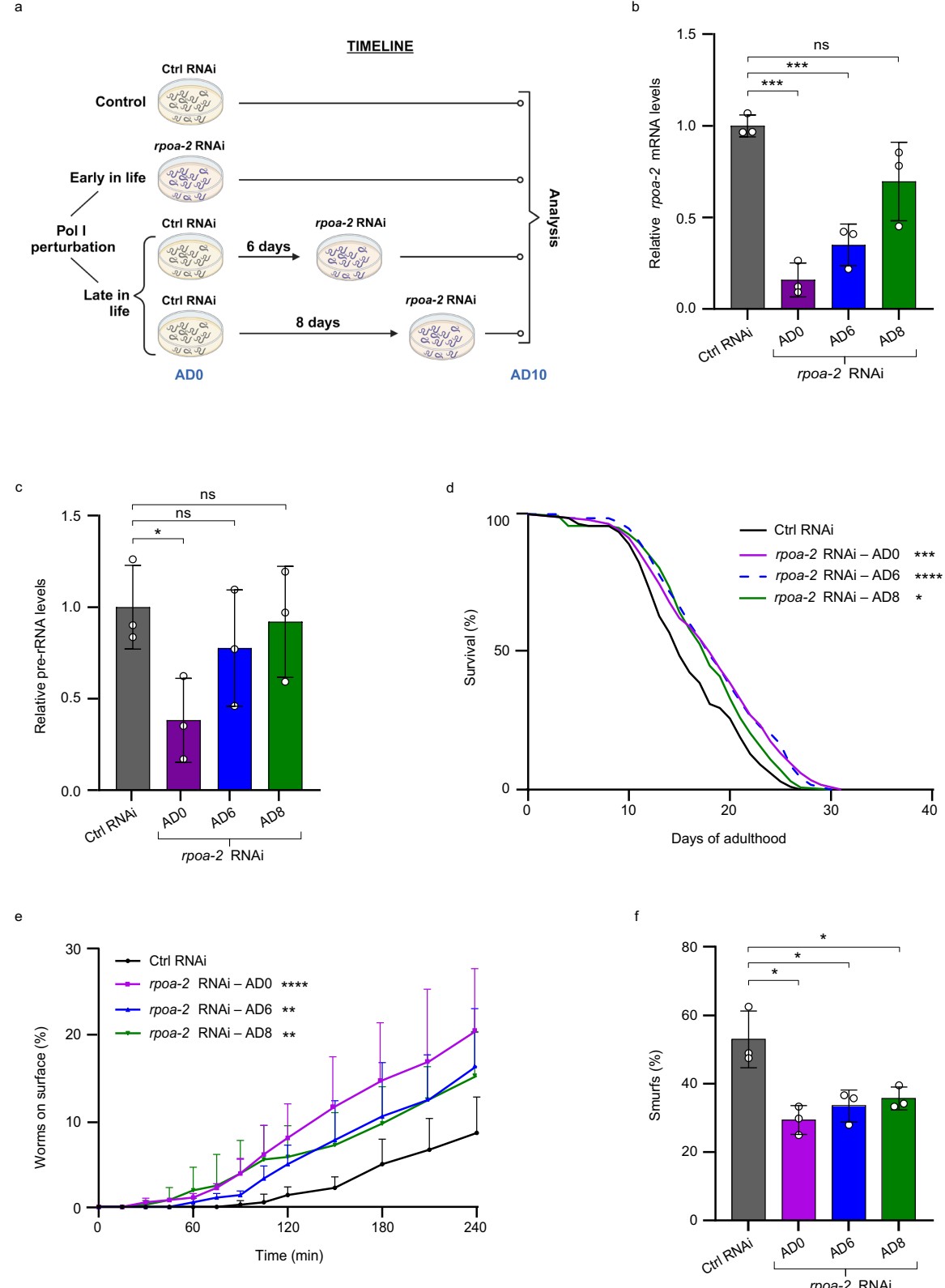

dampened Pol I activity preserves metabolic plasticity throughout life, as showcased by lowered mitochondrial stress at a young age and elevated late-life metformin resilience of worms with perturbed Pol I machinery. Concurrently, lowered Pol I activity triggers lipidome remodeling towards reduced abundance of damage-prone and lipo-toxic phospholipids and fatty acids, with increased Pol I activity con-ferring the opposite effect on both mitochondrial stress and the

lipidome. The most notable difference in lipidome profiles was the differential expenditure of cellular TAG storage, which was enhanced in *ncl-1* KD and reduced in *tif-1A* KD during aging. Remarkably, in both cases, the changes in lifespan could be reversed by the depletion of the lipase ATGL-1, which initiates TAG catabolism[51]. ATGL-1 is activated during fasting to facilitate the extraction of energy from TAGs[53] but at the same time, its activity is dampened during dauer diapause to ration

**Fig. 7 | Curbing of Pol I activity promotes healthy longevity independently of age. a** Schematic of the strategy for *rpoa-2* inactivation at different ages and subsequent analysis of the animals. The image was created with BioRender.com. **b, c** RT-qPCR analysis of *rpoa-2* mRNA (**b**) and pre-rRNA (**c**) levels at AD10 in nematodes treated with either control (Ctrl) RNAi or *rpoa-2* RNAi from AD0, AD6 or AD8 onwards. Data are relative to Ctrl RNAi values and normalized to *tgb-1* mRNA. Samples from 3 independent experiments with $n = 60$ animals per sample were analyzed, mean ± S.D. is shown. **d** Lifespan analysis of animals treated with Ctrl RNAi or with *rpoa-2* RNAi from AD0, AD6 or AD8 onwards. $n = 140$ worms, survival was scored daily. Result is representative of 3 independent tests. **e** Locomotor performance tests (burrowing assays) at AD10 after Ctrl RNAi or *rpoa-2* RNAi were

administered as shown in (**a**). Percentage of animals that have reached the surface within indicated time is shown. Data are from 3 independent trials with $n = 40$ worms per group, mean ± S.D. values are shown. **f** Quantification of 'Smurf' animals showing intestinal leakage after RNAi treatments as indicated in (**a**). Percentage of worms with whole body staining from $n = 40$–50 worms per treatment is presented. Mean ± S.D. values from 3 independent trials are shown. *P* values were determined with an unpaired Student's *t*-test (**b, c, f**) or Mantel-Cox test (**d, e**). ns = not significant, ****$P < 0.0001$, ***$P < 0.001$, **$P < 0.01$, *$P < 0.05$. All statistical tests used were two-sided, the exact *P* values and statistical analyses are reported in the Source Data file.

TAG usage during the exceptional longevity of the dauer larvae[52]. It is thus feasible that ATGL-1 activity is modulated by changes in energy levels downstream of differential Pol I output. In this context, *tif-1A* and *rpoa-2* KDs exhibited increased TAG levels, indicating slower TAG expenditure similar to the dynamics observed in dauer larvae, while TAG levels in *ncl-1* KD suggest accelerated TAG catabolism. Such differences in TAG mobilization are likely influencing the levels of FAs that are released from TAGs and further processed by mitochondria dictating oxygen consumption rate (OCR) and expression of mitochondrial homeostasis genes[57]. Consistently, we found OCR to be elevated in *ncl-1 KD* nematodes, while it was lowered in *tif-1A KD* animals. We also found that depletion of ATGL-1 reduces mitochondrial stress levels in the background of *ncl-1* KD. Given the connection between OCR and ROS production by the electron transport chain[58], our findings suggest that curbing of Pol I activity preserves mitochondrial health by attenuating TAG catabolism and thereby restricting OCR. Conversely, boosted rDNA transcription accelerates TAG turnover, in turn promoting oxygen consumption, damage of mitochondria, and premature aging (Fig. 6f and Supplementary Fig. 16). The premature demise of *ncl-1* KD worms delivers a note of caution for recreational self-treatments that activate TOR (e.g. high protein diets) and thereby boost Pol I activity. Such treatments do promote rapid improvements in fitness as seen in *ncl-1* deficient nematodes, but may accelerate metabolic aging in the long-term. Despite the diversity of observed lipidome changes, only TAG lipolysis was critical for the geroprotective effects of lowered rDNA transcription. It is thus plausible that at least some of the detected lipidome adaptations occurred as a result of altered TAG turnover. In line with this hypothesis, TAG cycling was linked to dynamic changes in the cellular FA repertoire[64], and increased accumulation of TAGs in lipid droplets was associated with the geroprotective lipidome remodeling in nematodes exposed to high levels of MUFA-lipids[65].

Importantly, we show that curbing of Pol I activity retains its geroprotective capacity also late in life, unlike DR and metformin treatment, which indirectly regulate Pol I[10–12]. It is conceivable that inhibition of pre-rRNA synthesis allows bypassing the adaptive failures that limit late life efficacy of metformin and DR[13,14] by acting directly on one of their prominent downstream targets. In addition, we provide evidence of metabolic stress protection by pharmacological Pol I inhibition in human primary cells. While our results raise hope for a new longevity treatment, interference with pre-rRNA synthesis in vivo would need to be carefully tuned. Pol I perturbation early in life comes with the risk of impaired development and growth retardation as seen in congenital diseases termed ribosomopathies[66]. Accordingly, our data suggest that initiating moderate Pol I inhibition in late life would bypass the potential developmental trade-offs and instead counteract the aberrant increase in rRNA production that has been reported in normal and pre-mature human aging[67].

## Methods
### *C. elegans* strains and maintenance
*C. elegans* strains were cultured under standard conditions as described in 'Wormbook'[68]. Age-synchronization of worms by bleaching was

performed as previously described[69]. Experimental temperatures for all figures are summarized in Supplementary Data 23. The *C. elegans* wild isolate (Bristol N2), *ife-2(ok306)* mutant (KX15), *atfs-1(gk3094)* mutant (VC3201), *pink-1(tm1779)* mutant (BR4006, without transgene), *skn-1(zj15)* mutant (QV225), *prx-5(ku517)* mutant (MH5239), *myo-3p::gfpmt (zcls14)* expressing strain (SJ4103), *hsp-6p::GFP* expressing strain (SJ4100) and FIB-1::GFP expressing strain (COP262) were obtained from the NIH funded Caenorhabditis Genetics Centre (CGC). *E. coli* OP50 and HT115 strains were obtained from CGC. The *tif-1A* overexpressing strain C0P2239 (genotype: *3p::3xTy1::C36E8.1::tbb-2u in ttTi5605, unc-119(+))* II; *unc 119(ed3)* III) was generated by InVivo Biosciences using the MosSCI method[70].

### RNA interference experiments
RNAi experiments were prepared as described elsewhere[71] using bacterial strains from the Ahringer RNAi feeding library (Source BioScience Ltd.). *ncl-1* and *tif-1A* knockdown was achieved by feeding the worms with bacteria producing the corresponding dsRNA for two generations at 15 °C. To achieve *rpoa-2* and *atgl-1* knockdown, L4 larvae or adult worms of the indicated ages were transferred on the dsRNA-producing bacteria and kept throughout the experiment at 20 °C, unless stated otherwise. RNAi treatment durations for all figures are summarized in Supplementary Data 23.

### Lifespan analysis in *C. elegans*
For lifespan analysis without RNAi treatment, age-synchronized worms were reared on nematode growth medium (NGM) agar with *E. coli* OP50 at 20 °C. If RNAi treatment was conducted during lifespan analysis, worms were kept at 15 or 20 °C as indicated in Supplementary Data 23, and NGM agar contained 1.5 mM Isopropyl-β-D-thiogalactopyranosid (IPTG, Sigma-Aldrich, I6758) and was seeded with a dsRNA-producing *E. coli* HT115(DE3) strain[72]. For each assay condition, two 60 mm plates with initially 70 worms each were used. Animals were transferred to fresh plates every second day during the reproductive phase and every 4 days in the post-reproductive phase. Worms were scored daily, and dead worms were removed. For lifespan analysis in the presence of drugs, cycloheximide (Sigma-Aldrich, 01810), Carbonyl cyanide-p-trifluoromethoxyphenylhydrazone (FCCP) (Cayman chemical, Cay15218-10) and metformin (M2009, TCI) were added to the agar to a final concentration of 10, 20, 25 μM and 50 mM, respectively. DMSO (Sigma-Aldrich, D2438) was used as a solvent for FCCP at a final concentration of 0.05%.

### Burrowing assay
The burrowing assay was performed as previously described[17]. Briefly, 50 μl of a 26% pluronic F-127 gel (Sigma-Aldrich, P2443) (w/w, dissolved in H2O) were solidified at the bottom of a 35 mm plate and 40-45 worms were placed on top. Animals were overlaid with 4 ml of the same gel and after hardening, 20 μl of concentrated *E. coli* OP50 suspension (100 mg/ml) was added as a chemoattractant on the surface. Animals reaching the surface were recorded every 15 or 30 min for 2 or 3 h. Three plates per condition were used in each experiment.

## Smurf assay

Smurf assay was performed following a previously described method[19]. In brief, nematodes were incubated in a *E. coli* OP50 liquid culture containing 5% Brilliant Blue FCF (Sigma-Aldrich, 80717) for 3 h at 18 °C. Afterward, worms exhibiting blue staining outside of the intestine (Smurf) were scored using Zeiss Axio Zoom.V16 microscope at the magnification of ×25.

## Nucleolar and worm size measurement

FIB-1::GFP expressing nematodes were washed with M9 buffer and transferred to poly-L-lysine-coated microscopy slides (Sigma-Aldrich, P5899). M9 buffer was removed and worms were fixed in 10 µl ethanol. After evaporation of the ethanol, fixation was repeated and slides were mounted with mounting media containing 4´,6-diamidino-2-phenylindole (DAPI). The specimens were imaged on Zeiss Axiovert 200 ApoTome (Carl Zeiss AG). Nuclei (DAPI staining, Abcam PLC, ab104139), and nucleoli (FIB-1::GFP staining) in individual cells were contoured and the corresponding areas were quantified using the ZEISS ZEN Imaging Software. For determining worm size, bright field images were recorded, and the areas of animals were measured with the same software.

## Gene expression analysis

For extraction of total RNA, worms were washed twice in M9 buffer, resuspended in TRI reagent (Sigma-Aldrich, T9424), and lysed with a Precellys tissue homogenizer (Bertin Technologies SAS). BJ cells were harvested and directly lysed in TRIzol (Thermo Fisher Scientific/Invitrogen, 15596018). Total RNA was isolated and 1 µg was treated with DNase I (Sigma-Aldrich/Merck, AMPD1-1KT) in the presence of 0.25 µl RNase inhibitor (RiboLock, Thermo Fisher Scientific, EO0384) for 20 min at room temperature. cDNA was synthesized with random hexamers (Roche, 11034731001) using SuperScript IV Reverse Transcriptase (Thermo Fisher Scientific, 18091200 and 18090050) according to the manufacturer's recommendation. The cDNA was amplified in a two-step quantitative PCR (qPCR) using SYBR Green Maxima SYBR Green/ROX 2X qPCR Master Mix (Thermo Fisher Scientific, K0222) following the manufacturer's instructions. QuantStudio™ 3 real-time-PCR instrument (Applied Biosystems, Foster City, USA) was used. Primer sequences are provided in Supplementary Data 22.

## ATP measurements

ATP content was measured as previously described[73] with minor modifications. Briefly, 50 worms were washed twice and resuspended in 1 ml M9 buffer, followed by snap-freezing and boiling for 10 min. After sonication in a Bioruptor Plus (Diagenode) (10 cycles, 1 min on, 30 s off), lysates were cleared by centrifugation at $15,300 \times g$ at 4 °C for 10 min. To determine relative ATP levels, the "ATP Bioluminescence Assay Kit HS II" (Roche, 11699709001) was used according to the manufacturer's instructions, and measurements were carried out in a Mithras LB 940 microplate reader (Berthold). Data were normalized to the protein concentrations of the samples as measured by absorbance at 280 nm (A280) in a NanoDrop photometer (Thermo Fisher Scientific).

## Western blot analysis

700 worms were lysed as for ATP measurements by snap freezing, boiling, and sonication. 30–50 µg of total protein were resolved by 10% polyacrylamide gel electrophoresis (PAGE). Proteins were transferred to a nitrocellulose membrane by semi-dry blotting on a Trans-Blot Turbo Transfer System (Bio-Rad Laboratories). Immunodetection was performed with primary anti-Ty1 (Diagenode, C15200054, Lot 007) or anti-tubulin (Sigma-Aldrich, clone DM1A, T6199) antibodies, followed by probing with a horseradish peroxidase-conjugated anti-mouse antibody (Dianova, DkxMu-003-FHRPX, Lot 67-83-071919).

Antibody dilutions were 1:5000 for primary, and 1:10,000 for secondary antibodies. Blots were developed with a SuperSignal west pico PLUS chemiluminescent substrate (Thermo Fisher Scientific, 34580) and imaged on a Fusion Solo 4S system (Vilber Lourmat).

## Sample preparation for proteomics

Proteomics analysis was conducted as previously described[14]. Briefly, 700 worms were harvested for each sample and transferred to Precellys CKMix Lysing Kits tubes containing 1.4 mm and 2.8 mm ceramic beads (Bertin Technologies). Samples were adjusted with 25 µl 10 x lysis buffer (final concentrations 4% SDS, 100 mM HEPES, pH 8.5, 50 mM DTT) and homogenized using a Precellys 24 tissue homogenizer (Bertin Technologies) for 3 cycles (6000 mixes for 20 s followed by 30 s pause). Beads were sedimented and supernatants were sonicated (Bioruptor Plus, Diagenode) for 10 cycles (30 s ON/60 s OFF) at 20 °C, followed by boiling at 95 °C for 5 min. Reduction was followed by alkylation with 20 mM iodoacetamide (IAA, final concentration 15 mM) for 30 min at room temperature in the dark. Protein amounts were estimated by SDS-PAGE and Coomassie blue staining of an aliquot of each sample by comparing the staining intensity to a quantified standard cell extract prepared from mouse embryonic fibroblasts. 30 µg protein was further processed for digestion. Proteins were precipitated with four volumes of ice-cold acetone overnight at −20 °C and sedimented by centrifugation at $20,800 \times g$ for 30 min at 4 °C, followed by two washes with 300 µL ice-cold 80% (v/v) acetone in water. Protein pellets were air-dried and resuspended by sonication in 25 µl of digestion buffer (1 M Guanidine, 100 mM HEPES, pH 8). LysC (Wako) was added at 1:100 (w/w) enzyme to protein ratio and digestion proceeded for 4 h at 37 °C. Samples were then diluted with one volume of MilliQ water and trypsin (Promega) was added at a 1:100 (w/w) enzyme:protein ratio. After digestion overnight at 37 °C, trifluoroacetic acid was added to a final concentration of 10% (v/v), and samples were desalted with Waters Oasis® HLB µElution Plate 30 µm (Waters Corporation) under a soft vacuum following the manufacturer's instruction. Samples were dried down using a speed vacuum centrifuge (Eppendorf Concentrator Plus) and redissolved at a concentration of 1 µg/µl peptides in reconstitution buffer (5% (v/v) acetonitrile, 0.1% (v/v) formic acid in MilliQ water).

## LC-MS/MS proteomics

Reconstituted peptides were spiked with the retention time normalization iRT kit (Biognosys) and separated in trap/elute mode using the nanoAcquity MClass Ultra-High Performance Liquid Chromatography system (Waters, Waters Corporation) equipped with a trapping (nanoAcquity Symmetry C18, 5 µm, 180 µm × 20 mm) and an analytical column (nanoAcquity BEH C18, 1.7 µm, 75 µm × 250 mm). The outlet of the analytical column was coupled directly to an Orbitrap Q exactive HF-X mass spectrometer (Thermo Fisher Scientific) using the Proxeon nanospray source. Solvent A was water and 0.1% formic acid, and solvent B was acetonitrile and 0.1% formic acid. The samples (~1 µg) were loaded with a constant flow of solvent A at 5 µl/min onto the trapping column. Trapping time was 6 min. Peptides were eluted via the analytical column with a constant flow of 0.3 µl/min. During the elution step, the percentage of solvent B increased in a nonlinear fashion from 0–40% in 120 min. Total run time was 145 min. The peptides were introduced into the mass spectrometer via a Pico-Tip Emitter 360-µm outer diameter × 20-µm inner diameter, 10-µm tip (New Objective) heated at 300 °C, and a spray voltage of 2.2 kV was applied. The capillary temperature was set at 300 °C. The radiofrequency ion funnel was set to 40%. For DIA data acquisition, full-scan mass spectrometry (MS) spectra with mass range 350–1650 $m/z$ were acquired in profile mode in the Orbitrap with a resolution of 120,000 FWHM. The default charge state was set to 3+. The filling time was set at a maximum of 60 ms with a limitation of $3 \times 10^6$ ions. DIA scans were acquired with

40 mass window segments of different widths across the MS1 mass range. Higher collisional dissociation fragmentation (stepped normalized collision energy; 25.5, 27, and 30%) was applied and MS/MS spectra were acquired with a resolution of 30,000 FWHM with a fixed first mass of 200 $m/z$ after accumulation of $3 \times 10^6$ ions or after filling time of 35 ms (whichever occurred first). Data were acquired in profile mode. For data acquisition and processing of the raw data, Xcalibur 4.0 (Thermo Fisher) and Tune version 2.9 were used.

## Proteomics data analysis

DIA raw data were analyzed using the directDIA pipeline in Spectronaut (v.13, Biognosysis AG). The data were searched against UniProt and a species-specific (*C. elegans*, 26.677 entries) Swissprot database. The data were searched with the following modifications: Carbamidomethyl (C) (Fixed) and Oxidation (M)/ Acetyl (Protein N-term) (Variable). A maximum of 2 missed cleavages for trypsin was allowed. The identifications were filtered to satisfy FDR of 1% on peptide and protein levels. Relative quantification was performed in Spectronaut for each paired and unpaired comparison using the replicate samples from each condition. The data (candidate table) and data reports (protein quantities) were then exported and further data analyses and visualization were performed with Rstudio using in-house pipelines and scripts[74]. Differently regulated proteins were analyzed for biological functions using WormCat[23].

## Lipidomics

For each lipidomics analysis, 700–800 age-synchronized worms were washed with M9 medium and shock-frozen in 100 μl medium. Liquid-liquid-extraction of lipids and subsequent targeted lipidomics was performed as reported[14,75]. In brief, glycerophospholipids (i.e., phosphatidylcholine, phosphatidylethanolamine, phosphatidylinositol, phosphatidylserine), glycerolipids (i.e., TAG) and free fatty acids were separated on an ACQUITY UPLC system (Waters) with a reversed-phase column (ACQUITY UPLC BEH C8; 1.7 μm, 2.1 × 100 mm; Waters) using previously established settings[14]. The LC system was coupled to a QTRAP 5500 mass spectrometer (Sciex) equipped with an electrospray ionization source, which was operated by Analyst 1.6.2 (Sciex). Lipids were measured by multiple or selected reaction monitoring in the negative (glycerophospholipids, free fatty acids) or positive ion mode (triacylglycerols)[14], and extracted chromatograms were processed using Analyst 1.6.3 (Sciex).

The absolute abundance of different lipid classes represents the summarized signal intensities of all analyzed lipid species of an indicated lipid class (for example, all analyzed PCs), which are normalized to the internal standard 1,2-dimyristoyl-sn-glycero-3-phosphocholine (DMPC; 850345P, Avanti Polar Lipids) as well as the number of worms. Proportions of individual lipids or lipid scores (e.g., TAGs with a unique total fatty acid chain length) were calculated as ratio of signal intensities (e.g., from individual TAG species) to the summarized total signal intensity of a given lipid class (e.g., TAGs).

## Cell culture

For MTT assays, BJ human skin fibroblasts (ATCC, #CRL-2522) were cultured in an incubator with a humidified atmosphere at 37 °C and 5% $CO_2$ as previously described[14]. Briefly, cells were grown in complete Dulbecco's modified eagle's medium (DMEM) (high glucose media, Sigma-Aldrich/Merck, D6429) supplemented with 10% of fetal bovine serum (FBS) (Sigma-Aldrich/Merck, F7524). Cells were seeded in 96 flat well plates at a density of 30,000 cells per well and incubated overnight prior to treatment. For the 24h-long pre-treatment with Pol I inhibitors or DMSO (Sigma Aldrich/Merck, D2438), complete DMEM was used. For metformin treatment, the media was replaced with glucose-free DMEM (Thermo Fischer Scientific, A1443001) supplemented with 10% FBS only. DMSO was used as solvent for the inhibitors at a final concentration of 0.01%.

For gene expression analysis, cells were maintained in StableCell™ DMEM (Sigma-Aldrich/Merck, D0819) high glucose, supplemented with 10% FBS (Capricorn Scientific, FBS-11A), 100 units/mL penicillin and 0.1 mg/mL streptomycin (Sigma-Aldrich /Merck, P4333). Cells were seeded and incubated overnight prior to the treatment with Pol I inhibitors or DMSO (Sima-Aldrich/Merck, Cat# D2650), which lasted 24 h.

## Mitochondrial activity assay

Filtered MTT solution (Sigma Aldrich/Merck, M5655) was added to cell cultures at a final concentration of 1 mg/mL. Plates were incubated for 3 h (37 °C, 5% $CO_2$) with light protection followed by media replacement with the MTT solubilization agent (0.1 N HCL, Roth X942.1 in absolute isopropanol, Roth 6752.3). Specific MTT absorbance was measured at 570 nm and background - at 690 nm, using a plate-reading spectrophotometer (Infinite M1000 Pro, Tecan). Metformin (M2009, TCI), BMH-21 (SML1183, Merck) and CX-5461 (HY-13323A, MCE) were used for treatment.

## Assessment of mitochondrial stress and mitochondrial morphology

*hsp-6p::GFP* worms were age-synchronized and fed with empty vector (Ctrl), *mip-1* RNAi, *tif-1A* RNAi, or *ncl-1* RNAi bacteria (Ahringer library) from L4 stage of the previous generation. F1 adults were bleached and F2 progeny were used for the imaging on adulthood day 2 (AD2) using AxioZoom_V16 microscope (Zeiss) at ×90 magnification. Fluorescence was measured as the mean intensity value obtained from the GFP channel for each worm using the Zen blue (2.6 edition) software. For morphology tests, *myo-3p::gfpmt* expressing worms were fed with empty vector (Ctrl) RNAi from L1 stage and transferred to Ctrl, *rpoa-2* RNAi, or *ncl-1* RNAi at L4 stage, and % of tubular, intermediate, fragmented and very fragmented mitochondria was assessed on day 2 of adulthood as previously described[14].

## Assessment of mitochondrial OCR by Seahorse

Wild-type nematodes were age-synchronized and fed with empty vector (Ctrl), *tif-1A* RNAi or *ncl-1* RNAi bacteria (Ahringer library) from L4 stage of the F0 generation at 20 °C. F1 adults were bleached and F2 progeny were shifted to 15 °C at L4 stage and later used for the OCR measurement on day 12 of adulthood. Worms were rinsed off the plates and washed three times with 1X M9 buffer to remove traces of bacteria. Worms were then suspended in 1 ml EPA water and transferred to individual wells of a 96 well plate (20–30 worms/well). FCCP (Cayman chemical, Cay15218-10) and Rotenone (Sigma Aldrich/Merck, R8875) were dissolved in DMSO (Sigma Aldrich/Merck D2438) prior to assessment and diluted in EPA water to obtain the working concentration of 200 μM and 50 μM, respectively. The assay was performed in 96-well plate in Agilent Seahorse XF Pro analyser. Data was normalized to the number of worms.

## Statistical analysis

Experiments were performed in biological replicates as stated in the figure legends. Statistical analysis was performed with Prism 8 (Graphapd Software Inc.). All bar graphs show the mean values with error bars (±S.D. or ±S.E.M as indicated in the legends). For statistics, significance was determined at $P < 0.05$. The appropriate statistical test was chosen for each experiment as indicated in the figure legends and the Source Data file.

## Reporting summary

Further information on research design is available in the Nature Portfolio Reporting Summary linked to this article.

## Data availability

The mass spectrometry proteomics data, including the exact UniProt information, have been deposited to the ProteomeXchange

Consortium via the PRIDE[76] partner repository with the dataset identifier PXD028600. The mass spectrometry lipidomics data generated in this study have been deposited in the Metabolomics Workbench database (an international repository for metabolomics data and metadata, metabolite standards, protocols, tutorials and training, and analysis tools[77]) under Project ID PR001490 (https://doi.org/10.21228/M84D89). Source data generated in this study are provided in Supplementary Data and in the Source Data file. Respective data references can be found in the text and Figure legends. Source data are provided with this paper.

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

## Acknowledgements

We thank Lilia Espada for her important experimental input at the earlier stages of this study, and Melike Bayar for the valuable support at the final steps. We thank the Proteomics Core Facility and SPARK Technology Transfer Core Facility at FLI for supporting this study. The FLI is a member of the Leibniz Association and is financially supported by the Federal Government of Germany and the State of Thuringia. H.B., M.E., S.S., L.L. and P.C. were supported by the EU-ESF Thüringer Aufbaubank funding (2019 FGR 0082). HB, SS and LL received funding from the Thuringian state program ProExzellenz (RegenerAging—FSU-I-03/14) of the Thuringian Ministry for Research. M.E., H.B., O.W., A.E. and E.O. are funded by the Carl-Zeiss-Stiftung via IMPULS consortium. M.E. is supported by the ERC CoG LifeLongFit of the European Commission and is a member of Cluster of Excellence Balance of the Microverse funded by the Deutsche Forschungsgemeinschaft (DFG). A.K. is supported by the Phospholipid Research Center (AKO-2019-070/2-1) and the Austrian Science Fund (P 36299). *C. elegans* strains were obtained from the Caenorhabditis Genetics Center (CGC), which is funded by NIH.

## Author contributions

H.B. and M.E. conceptualized and designed the study; S.S., P.C., A.M., A.E., L.L., Y.W., E.O. and H.L. performed experiments; H.B. and M.E. supervised data analysis and visualization; M.E., S.S., P.C., A.M., A.E., L.L., Y.W., E.O. and H.L. analyzed the data; N.R. performed sample curation for proteomics analysis; J.K. and A.O. developed proteomics analysis protocols and analyzed proteomics data; A.K. designed the lipidomics

study; F.W., A.G. and A.K. performed the lipidomics analysis; A.K., O.W., H.B. and M.E. interpreted the lipidomics data; S.S., P.C. and M.E. prepared figures; S.S., P.C. and M.E. performed statistical analysis; H.B. wrote the initial draft of the manuscript, M.E. revised the manuscript and H.B., S.S., P.C., A.K., O.W. and A.O. reviewed the manuscript.

## Funding

## Competing interests

The authors declare no competing interests.
