## [Peer Review File · Nature Communications]

Reducing the metabolic burden of rRNA synthesis promotes healthy longevityREVIEWER COMMENTS

Reviewer #1 (Remarks to the Author):

In general, this manuscript described a good story about the relationship between Pol I activity and lifespan. DIA-based proteomic analysis was applied to monitor age- and rDNA activity-dependent proteome changes in *Caenorhabditis elegans*. The experiments were well-designed and results were interesting. However, there are several key issues about proteomic data in this manuscript, as mentioned below, that should be clarified or supplied before the acceptance of manuscript.

Major concerns

- 1) Though DIA-based proteomics have performed for nine samples in this study, there is no information about the number of protein groups identified and quantified for each sample. In the supplemental tables, only data for selected protein groups have been provided.
- 2) Though the authors have performed a comprehensive proteomic study, however, they didn't analyze the proteomic data to get a whole view of proteomic changes in different conditions, instead, they only gave specific focus on adaptive responses previously linked to longevity, such as immune-, oxidative stress and proteostasis pathways, and only analyzed the expression of a limited number of proteins. Are there any other proteins that display age- and rDNA activity-dependent changes in the treatments and have some relationship with longevity?
- 3) In the supplemental table S1 and S2, some CLEC-, HSP-, and GST-related proteins displayed significantly changes in different treatments, for example, in TIF-1A vs EV2 D2 CELC section, ratio for G5EBG4 is 0.298, ratio for Q4W528 is 4.526. For the view of individual proteins, protein expression of these proteins has changed in the treatments, what are the relationships between expression changes of these proteins and the observed functional changes?

Minor points

- 1) For Extended data Fig.2, the authors declared that "a clear separation between age groups and RNAi treatments, with both tif-1A KD and ncl-1 KD animals clustering apart from controls at all ages". However, from my point of view, the PCA analysis is not very good, as

the three control groups have some overlap with the three ncl-1 RNAi groups.

2)The authors used lysis buffer containing both SDS and DTT to extract proteins from *Caenorhabditis elegans*. This lysis buffer is not compatible with the commonly-used protein assays (BCA or Bradford method) to measure protein concentration. So, the authors estimated protein amounts by SDS-PAGE and Coomassie blue staining. I am curious how to estimate protein amounts with this method?

Reviewer #2 (Remarks to the Author):

This paper found that altering pre-rRNA synthesis affects lifespan, which is linked to changes in mitochondrial function and lipid metabolism. Reducing Pol I activity, which regulates pre-rRNA synthesis, extends lifespan and improves healthspan in *C. elegans*. This longevity effect appears to be conserved in both fly and human cells. Moreover, differential Pol I activity regulates mitochondrial function and lipid metabolism, according to proteome and lipidome analysis. These changes may contribute to longevity. While the link between rRNA synthesis (Pol I activity) and aging is intriguing, it's unclear whether this effect is similar to that of inhibiting ribosome proteins, which has been shown to extend lifespan. Furthermore, while mitochondrial and metabolic changes are associated with longevity in nematodes with differential Pol I activities, it's unclear whether these changes are responsible for the increased lifespan.

Specific comments:

1. In worms, knocking down ribosome proteins (such as rps-15 and rps-22) increases their lifespan. Is it possible that the lifespan-extending effect observed in this manuscript is similar to published results of ribosome proteins, because both ribosome proteins and rRNA affect ribosome function? Or do ribosome proteins and rRNA regulate lifespan in different ways? This is a crucial point because it affects the paper's overall novelty.
2. Inhibition of ribosome proteins and translation-initiation factors regulate lifespan in different ways, according to a previous study [PMID: 17266679]. For example, knocking down translation-initiation factors requires DAF-16 to increase lifespan, whereas ribosome proteins do not. Does tif-1A RNAi require these factors to extend lifespan?
3. Ncl-1 RNAi increases hsp-6 slightly; how about tif-1A RNAi? It's possible that both

increasing and decreasing rRNA synthesis put mitochondria under stress.

4. While ATP levels correlate with lifespan, it's unclear whether preserving ATP indeed extends lifespan in the context of altered rRNA synthesis. Additionally, as ribosome is also important for mitochondria biogenesis, is mitochondrial biogenesis affected by ncl-1 or tif-1A RNAi? It's also important to examine mitochondrial morphology.

5. The authors provided limited information on lipid changes in Fig 5. I'm not sure why they only show TAG fatty acids with three or more double bonds, and MUFAs/PUFAs in PC and PE. Because both MUFA and some PUFAs have been shown to increase lifespan, the levels of MUFAs and major PUFAs should be displayed separately, rather than just as a ratio or n>3 double bonds.

6. Other lipid parameters, such as PC/PE ratio and UFA, SFA, and UFA/SFA (unsaturated FA/saturated FA) in PC and PE fractions, should be calculated. These parameters are all important regulators of membrane homeostasis and may also influence lifespan.

7. Because many fatty acids and membrane lipids, such as PC, have been shown to influence *C. elegans* lifespan, the authors should investigate whether they are involved in lifespan regulation in their system.

8. Some of the control lifespan data differ from one another. In comparison to the data in the other figures, the lifespan of the control RNAi group is very short in fig 6d. Is there a reason for this? In addition, the authors should include a separate table that shows the repeats of each lifespan data.

Reviewer #3 (Remarks to the Author):

The manuscript by Sharifi and colleagues examines the impact of ribosomal RNA synthesis on ageing. The authors use genetic interventions to alter the biogenesis of rRNA, principally targeting the RNA polymerase I machinery, to show that rRNA biogenesis limits longevity in worms and flies. The authors then perform proteomic and lipidomic studies to examine the underlying molecular mechanisms. The authors find that reduced protein synthesis is not likely to account for all benefits of reduced rRNA biogenesis. They present evidence that metabolic changes resulting in improved energy homeostasis are important mediators of the longevity observed upon reduction in rRNA biogenesis.

I find the manuscript interesting, addressing an important question, but with some major and some minor limitations. I would like the authors to address these before I could recommend the manuscript for publication in Nature Communications.

One issue is novelty. Previous work has examined how limiting RNA polymerase I activity impacts longevity: Tiku et al 2017 presented an albeit limited assessment of *tif-1a* knockdown in worms and found no or very little effect on lifespan, Corrales et al 2020 showed that partially inhibiting RNA polymerase I extended lifespan in the fruit fly. The manuscript by Sharifi takes this forward in that the authors further characterise the effects of reducing rRNA biogenesis to claim the longevity is due to metabolic changes. However, most of the findings are correlational and remain at the level of describing the phenotype of the long-lived animals. I think that direct demonstration that the longevity is due to metabolic changes would substantially strengthen the manuscript. This could be done in a way similar to examining the relationship between translation, rRNA biogenesis and longevity in the current manuscript (see e.g. Extended Data Figure 5).

Additionally, the authors should make clear note of the above-mentioned work in the introduction (the work is cited but not explained in sufficient detail) and include discussion of any differences in findings.

Furthermore:

For both 'omics analyses, p values have to be adjusted to account for multiple comparisons. Please make sure this is clear in the text. Details of the computational analyses performed should be given, especially regarding significance testing and multiple testing corrections. I would expect the lipidomics dataset to also be deposited in a public database e.g. Metabolights as well.

Fig 2a and b – there should be controls for each transgenic construct alone as each insertion can have an effect i.e. *tub-GAL80* and *UAS* controls are missing. Only 15 animals per condition? This does not sound correct. Additionally, please make explicit in the text that this lifespan was performed at high temperature. The transgenes do not appear to have been backcrossed to a common genetic background. They need to be.

Page 4 lines 20-21, the elevated pre-rRNA in ncl-1 kd is more likely to be due to accumulation of pre-rRNA due to an impairment in processing than due to increased pre-rRNA transcription, is it not?

Reviewer #4 (Remarks to the Author):

In this manuscript, the authors explored the effects of perturbation of pre-rRNA synthesis on longevity in *C.elegans* and *Drosophila* and identified that reduced pre-rRNA synthesis can improve lifespan through decrease of protein synthesis, preservation of energy and modulation of lipid metabolism. The data is in contrast to various studies in yeast and human cells have shown that rDNA instability and inhibition of Pol I activity can induce aging and senescence, respectively (Sharifi et al Annual review of Biochemistry, 2018; Hein et al the nucleolus and ribosome genes in aging and senescence, 2012).

Major concerns:

In Figure 4 d-e, the authors suggest that “Pol I inhibition by BMH-21 and CX-5461 compounds, promoted human cell survival under metabolic stress”. I find the data perplexing for the following reasons: 1) Inhibition of Pol I transcription has been shown to induce senescence in human fibroblasts (Quin et al 2016); 2) MTT assay may not be a reliable method for assessing cell survival if the effect of drug treatment is cell cycle arrest and senescence, 3) It seems that the combination effect of CX-5461/metformin and BMH-21/metformin is normalised to single agent CX-5461 and BMH-21 alone effects. If this is the case, data interpretation is misleading as it is likely masking the inhibitory effects of the Pol I inhibitors on cell proliferation. 4) Metformin has been shown to sensitise MYC-driven lymphoma cells to CX-5461/ everolimus therapy in vivo Kusnadi et al., EMBO J 2020). The authors should discuss their findings in relation to previous research in various model systems that demonstrated the growth inhibitory effects of Pol inhibition.

In fact, the manuscript ignores a lot of work in mammalian model systems that show defects in Pol transcription leads to the induction of a nucleolar stress response and growth inhibitory phenotypes. Defects in ribosome biogenesis, including for example impaired pre-rRNA synthesis due to mutations in TCOF1, have been causatively linked to the pathogenesis

of a group of diseases called ribosomopathies. Clinical manifestations of ribosomopathies include growth retardation and pre-mature aging (Hannan et al BBA 2013).

Additional comments:

In figure 2, the authors show reduced body size and neuromuscular performance in *tif-1A* deficient worms at day 2 of adulthood. What are the effects of *tif-1A* deficiency on the worm development at younger age? P53 activation plays a critical role in the pathogenesis of ribosomopathies due to activation of a nucleolar stress response. Does the knockdown of *tif-1A* affect p53 expression/activity in the model systems used in this study?

Figure 1a-f showed that modulation of *tif-1A* expression level affected the lifespan. It is not clear at what stage (young, middle or old age) *tif-1A* expression was modulated and whether this effect is independent on the stage at which pol I perturbation occurs.

The authors showed a significant downregulation of global protein synthesis in *tif-1A* deficient worms. Interestingly, *tif-1A* knockdown promoted a moderate but significant elevation of mito-ribosome levels at old age. This suggests that selective translational adaption may underly longevity under conditions of reduced rRNA synthesis. The authors can further analyze and identify proteins with upregulated expression levels in the proteomics data and the biological processes involved in this process.

Mitochondrial activity and function are associated with longevity and aging (Lima et al, Nature Aging 2, 199-213, 2022). While the authors showed *tif-1A* depletion preserved the mito-ribosome levels and the ATP content in late life, there is no direct evidence to support maintenance of mitochondrial activity in *tif-1A* deficient worms. It is critical to measure mitochondrial oxidative phosphorylation by seahorse mitochondrial stress test to assess mitochondrial activity in *tif-1A* deficient worms.

We would like to thank all reviewers for their insightful comments, which helped us improve our manuscript and enhance the clarity of data presentation. Responses to the specific comments can be seen below. The revised parts of the manuscript text are marked with the grey highlight.

Reviewer 1.

In general, this manuscript described a good story about the relationship between Pol I activity and lifespan. DIA-based proteomic analysis was applied to monitor age- and rDNA activity-dependent proteome changes in *Caenorhabditis elegans*. The experiments were well-designed and results were interesting. However, there are several key issues about proteomic data in this manuscript, as mentioned below, that should be clarified or supplied before the acceptance of manuscript.

Major concerns

1) Though DIA-based proteomics have been performed for nine samples in this study, there is no information about the number of protein groups identified and quantified for each sample. In the supplemental tables, only data for selected protein groups have been provided.

We thank the reviewer for their suggestion to improve the presentation of the proteomics data. The distribution of quantified protein numbers in all measured samples is now depicted in the Supplementary Fig. 2, and complete data for all comparisons used in the figures is provided in Table S1. The entire proteomics data set is deposited to the ProteomeXchange Consortium via the PRIDE partner repository with the dataset identifier PXD028600, and information about the availability of these data is provided on page 21, line 11 of the revised manuscript.

2) Though the authors have performed a comprehensive proteomic study, however, they didn't analyze the proteomic data to get a whole view of proteomic changes in different conditions, instead, they only gave specific focus on adaptive responses previously linked to longevity, such as immune-, oxidative stress and proteostasis pathways, and only analyzed the expression of a limited number of proteins. Are there any other proteins that display age- and rDNA activity-dependent changes in the treatments and have some relationship with longevity?

We indeed performed a literature-driven targeted analysis of longevity-associated pathways in Fig. 3a, b. However, an unbiased analysis of the EV, *tif-1A* KD and *ncl-1* KD proteomes was also carried out using the WormCat gene set enrichment analysis tool (Holdorf et al., 2020). This analysis is presented in Fig. 3c and d, Supplementary Fig. 4 and 5, and Tables S4-S9, and it revealed strong enrichment of mitochondrial and lipid metabolism pathways among the proteins differentially expressed in response to modulation of Pol I activity. This finding was followed up experimentally in the next figures leading to key mechanistic conclusions of this study. In response to the reviewer's comment, we have now modified the text (page 7, lines 12-18; page 8, lines 3-5) to clarify our dual approach to proteome analysis (targeted and unbiased).

3) In the supplemental table S1 and S2, some CLEC-, HSP-, and GST-related proteins displayed significantly changes in different treatments, for example, in TIF-1A vs EV2 D2 CELC section, ratio for G5EBG4 is 0.298, ratio for Q4W528 is 4.526. For the view of individual proteins, protein expression of these proteins has changed in the treatments, what are the relationships between expression changes of these proteins and the observed functional changes?

The sets of proteins highlighted by the reviewer (CLECs, HSPs, and GSTs) are used as markers of cellular stress responses (immune response, proteostasis response and oxidative stress response respectively) in *C. elegans* (Espada et al., 2020). When a given response is triggered, the entire set of relevant markers (e.g. the entire group of CLECs) is expected to exhibit a significant change as shown previously (Ermolaeva et al., 2013; Espada et al., 2020). However, such coordinated changes were not observed in the analysis shown in Fig. 3a and b and Tables S2 and S3, leading to the conclusion that the evaluated stress responses are not triggered by modulation of Pol I activity. We agree with the reviewer that the use of designated proteins as stress markers needs to be explained more explicitly including the acknowledgement of changes of individual marker components, and we now modified the manuscript text accordingly (page 7, lines 12-18).

Minor points

1) For Extended data Fig.2, the authors declared that “a clear separation between age groups and RNAi treatments, with both *tif-1A* KD and *ncl-1* KD animals clustering apart from controls at all

ages". However, from my point of view, the PCA analysis is not very good, as the three control groups have some overlap with the three *ncl-1* RNAi groups.

We thank the reviewer for pointing out the lack of clarity in this part. The text was modified to acknowledge the partial overlap between control and *ncl-1* KD proteomes (page 7, lines 3-9). The overall conclusion (e.g. opposite trends between *tif-1A* KD and *ncl-1* KD in clustering away from the control) remains unaffected by this clarification.

2) The authors used lysis buffer containing both SDS and DTT to extract proteins from *Caenorhabditis elegans*. This lysis buffer is not compatible with the commonly-used protein assays (BCA or Bradford method) to measure protein concentration. So, the authors estimated protein amounts by SDS-PAGE and Coomassie blue staining. I am curious how to estimate protein amounts with this method?

In response to this comment, the clarification of the protein measurement step has been added to the methods section of the manuscript (methods file, page 5, lines 8-11).

Reviewer 2

This paper found that altering pre-rRNA synthesis affects lifespan, which is linked to changes in mitochondrial function and lipid metabolism. Reducing Pol I activity, which regulates pre-rRNA synthesis, extends lifespan and improves healthspan in *C. elegans*. This longevity effect appears to be conserved in both fly and human cells. Moreover, differential Pol I activity regulates mitochondrial function and lipid metabolism, according to proteome and lipidome analysis. These changes may contribute to longevity. While the link between rRNA synthesis (Pol I activity) and aging is intriguing, it's unclear whether this effect is similar to that of inhibiting ribosome proteins, which has been shown to extend lifespan. Furthermore, while mitochondrial and metabolic changes are associated with longevity in nematodes with differential Pol I activities, it's unclear whether these changes are responsible for the increased lifespan.

Specific comments:

1. In worms, knocking down ribosome proteins (such as *rps-15* and *rps-22*) increases their lifespan. Is it possible that the lifespan-extending effect observed in this manuscript is similar to

published results of ribosome proteins, because both ribosome proteins and rRNA affect ribosome function? Or do ribosome proteins and rRNA regulate lifespan in different ways? This is a crucial point because it affects the paper's overall novelty.

The reviewer is right and lifespan extension upon knock down of ribosomal proteins has previously been described as well as longevity enhancement by Pol I inactivation, and both are expected to influence ribosome biogenesis. In this context, the outstanding contribution of our study is in demonstrating the prevailing role of energy metabolism in connecting dampened pre-rRNA transcription but not impaired translation to enhanced longevity (Fig. 4a and b, and Supplementary Fig. 6b), and these findings are entirely new. To test if downregulation of ribosomal proteins elicits energy saving similar to dampened pre-rRNA transcription, we now measured ATP levels in control and *rps-15* KD worms. The result of this experiment (Supplementary Fig. 6c) demonstrated that metabolism is a common downstream target of distinct processes implicated in ribosome biogenesis expanding our novel conclusions to a broader range of longevity modifiers (page 10, lines 10-16). We thank the reviewer for pointing us towards this additional test, which enhanced the conceptual output of our study (page 12, lines 5-9).

2. Inhibition of ribosome proteins and translation-initiation factors regulate lifespan in different ways, according to a previous study [PMID: 17266679]. For example, knocking down translation-initiation factors requires DAF-16 to increase lifespan, whereas ribosome proteins do not. Does *tif-1A* RNAi require these factors to extend lifespan?

The data presented in Fig. 4a, b and Supplementary Fig. 6b, c indeed suggest that the novel metabolic link uncovered by our study holds true for interventions that affect ribosomal biogenesis but not translation at the level of initiation or elongation. In line with longevity mechanisms downstream of Pol I repression and translational inhibition being distinct, the adaptive stress responses such as immune, proteostasis and oxidative stress response known to be DAF-16 targets (Hsu et al., 2003; Murphy et al., 2003) are not upregulated in *tif-1A* or *ncl-1* KD worms (Fig. 3a and b) and stress response pathways co-regulated by DAF-16 (mitophagy and mitochondrial UPR (Princz et al., 2020; Wu et al., 2018b) are not required for the longevity extension by Pol I inactivation (Fig. 6c and d). In the context of the previous literature (Hansen

et al., 2007), our data supports the model that interference with ribosome biogenesis mainly extends longevity by energy saving and remodeling of energy metabolism with no significant engagement of the adaptive stress responses including the ones regulated by DAF-16. To better embed our findings in the context of previous reports, we now added the comparative reflection of our own and earlier discoveries to the discussion section of the manuscript (page 18, lines 10-24).

3. *Ncl-1* RNAi increases *hsp-6* slightly; how about *tif-1A* RNAi? It's possible that both increasing and decreasing rRNA synthesis put mitochondria under stress.

In response to this comment, we tested the expression of *hsp-6 GFP* reporter in *tif-1A* KD animals and found it to be reduced below control levels (Fig. 6a, right panel), indicative of the lowered mitochondrial stress (mito-stress) in these animals, which is consistent with the preservation of the mitochondrial function by *tif-1A* KD during aging shown in Fig. 4d and the extended lifespan of these worms (page 15, lines 9-14).

4. While ATP levels correlate with lifespan, it's unclear whether preserving ATP indeed extends lifespan in the context of altered rRNA synthesis. Additionally, as ribosome is also important for mitochondria biogenesis, is mitochondrial biogenesis affected by *ncl-1* or *tif-1A* RNAi? It's also important to examine mitochondrial morphology.

We thank the reviewer for pointing out the lack of clarity in the discussion of the energy saving response to Pol I inhibition: while we detect the reduction of ATP expenditure as one of the clearest metabolic consequences of Pol I inactivation, we do not have evidence to suggest that it is causal to the other remodeling effects on metabolism, which we observed. To accommodate this important point, we re-phrased the chapter title on page 9, line 2-3.

To test the involvement of mitochondrial biogenesis in the longevity effects of Pol I inactivation, we now combined Pol I inhibition by *rpoa-2* RNAi with the loss of function mutation in biogenesis regulator *skn-1/Nrf2* (Palikaras et al., 2015). Importantly, Pol I KD failed to extend longevity in the absence of *skn-1* (Fig. 6f) demonstrating the key role of mitochondrial biogenesis in mediating metabolic preservation of Pol I inactivated animals. We thank the

reviewer for pointing us towards this important experiment that significantly enhanced the conclusions of our study (page 16, lines 6-16).

To respond to this comment, we also analyzed mitochondrial morphology in *ncl-1* KD and Pol I inactivated animals observing increased fragmentation of mitochondria in *ncl-1* inactivated worms (Fig. 6b and Supplementary Fig. 10b), consistent with increased levels of mitochondrial stress in this background seen by *hsp-6 GFP* expression tests (Fig 6a). The slight elevation of fission observed in *rpoa-2* KD worms (Fig. 6b and Supplementary Fig. 10b), which was not accompanied by UPR^{mt} induction (Fig. 6a), likely reflects homeostatic remodeling of the mitochondrial network and is consistent with the involvement of mitochondrial biogenesis in the longevity extension downstream of Pol I (page 15, lines 14-21).

5. The authors provided limited information on lipid changes in Fig 5. I'm not sure why they only show TAG fatty acids with three or more double bonds, and MUFAs/PUFAs in PC and PE. Because both MUFA and some PUFAs have been shown to increase lifespan, the levels of MUFAs and major PUFAs should be displayed separately, rather than just as a ratio or $n \geq 3$ double bonds.

We thank the reviewer for pointing out the lack of clarity in describing our initial lipidome analysis approach: long chain TAGs and TAGs with $n \geq 3$ PUFAs were investigated as previously described markers of mitochondrial dysfunction during aging (Espada et al., 2020) observing a reduction of both markers in *tif-1A* KD and elevation in *ncl-1* KD in line with the opposite longevity changes and metabolic stability of the KDs. MUFA/PUFA ratios in PCs and PEs were previously chosen because they show biggest differences between genetic backgrounds, however, in response to this comment we now display MUFA/PUFA ratios in all lipid classes (Supplementary Fig. 9a and Table S17). Levels of individual MUFAs and PUFAs are now also displayed in Supplementary Fig. 8a and Table S16, and manuscript text is adjusted accordingly (page 13, line 21 – page 15, line 2). The key conclusions are further supported by the newly displayed data.

6. Other lipid parameters, such as PC/PE ratio and UFA, SFA, and UFA/SFA (unsaturated FA/saturated FA) in PC and PE fractions, should be calculated. These parameters are all important regulators of membrane homeostasis and may also influence lifespan.

The relative levels of membrane phospholipids including phosphatidylcholines (PC) and phosphatidylethanolamines (PE) are now displayed in Supplementary Figure 8b-g and Table S16, showing increased levels of PCs and decreased levels of PEs in *tif-1A* KD animals while *ncl-1* KD shows opposite trends. As PEs are more prone to peroxidation compared to PCs, this result is consistent with opposite prevalence of metabolic stress observed in Pol I inactivated (lower stress) and boosted (higher stress) worms. The relative abundance of saturated fatty acids (SFAs), PUFAs and MUFAs in *tif-1A* KD and *ncl-1* KD worms is displayed in Fig. 5d and Table S13 showing increased abundance of MUFAs and reduced levels of SFAs and PUFAs in *tif-1A* KD worms. As MUFAs are less prone to peroxidation than PUFAs, and SFAs contribute to lipotoxicity (Mutlu et al., 2021), the fatty acid profile of Pol I inactivated animals is also consistent with their increased longevity, while short-lived and metabolically stressed *ncl-1* KD worms display an opposite fatty acid profile. The distribution of distinct fatty acids including SFAs and UFAs among lipid species is additionally visualized in Supplementary Fig. 8a and Table S16.

7. Because many fatty acids and membrane lipids, such as PC, have been shown to influence *C. elegans* lifespan, the authors should investigate whether they are involved in lifespan regulation in their system.

The contribution of individual lipids to differential longevity of Pol I inactivated and boosted animals is outside the scope of this study, which focuses on uncovering a general principle that impairment of ribosome biogenesis regulates longevity via metabolic remodeling and not only via dampening of translation. We successfully used unbiased lipidome profiling as a tool for characterizing the overall metabolic state of Pol I inactivated and boosted animals, and the rich resource of the uncovered specific changes can be employed by future follow up studies to determine the possible role of distinct lipids or fatty acids in life extension following reduced rDNA activity. However, several findings in proteomics and lipidomics data (Fig 3a, b and Fig 5a) suggested that preservation and mobilization of TAGs is possibly implicated in Pol I regulated longevity. To test this possibility we combined Pol I inactivation by *rpoa-2* KD with the KD of the key TAG mobilization enzyme ATGL-1. Importantly, single knock down of *atgl-1* conferred life extension (Fig 5b) in line with the TAG preservation being positively correlated with longevity as seen in Fig 5a. Concurrently, *atgl-1* KD abrogated the capacity of *rpoa-2*

RNAi to trigger lifespan extension (Fig. 5b), demonstrating equal importance of TAG preservation and mobilization for the longevity benefits seen in Pol I inactivated animals (page 13, lines 7-20).

8. Some of the control lifespan data differ from one another. In comparison to the data in the other figures, the lifespan of the control RNAi group is very short in fig 6d. Is there a reason for this? In addition, the authors should include a separate table that shows the repeats of each lifespan data.

For lifespan measurements involving *tif-1A* and *ncl-1* RNAi, worms have been grown at 15 °C, while lifespan experiments involving *rpoa-2* KD (e.g. Fig. 6d) were carried out at 20 °C because of the late onset of RNAi exposure. This information has been provided in the methods section, and we have now included it also in the figure legends. Additionally, we now provide information about all repeats of the lifespan experiments in a dedicated Survival data file.

Reviewer 3.

The manuscript by Sharifi and colleagues examines the impact of ribosomal RNA synthesis on ageing. The authors use genetic interventions to alter the biogenesis of rRNA, principally targeting the RNA polymerase I machinery, to show that rRNA biogenesis limits longevity in worms and flies. The authors then perform proteomic and lipidomic studies to examine the underlying molecular mechanisms. The authors find that reduced protein synthesis is not likely to account for all benefits of reduced rRNA biogenesis. They present evidence that metabolic changes resulting in improved energy homeostasis are important mediators of the longevity observed upon reduction in rRNA biogenesis.

I find the manuscript interesting, addressing an important question, but with some major and some minor limitations. I would like the authors to address these before I could recommend the manuscript for publication in Nature Communications.

One issue is novelty. Previous work has examined how limiting RNA polymerase I activity impacts longevity: Tiku et al 2017 presented an albeit limited assessment of *tif-1a* knockdown in

worms and found no or very little effect on lifespan, Corrales et al 2020 showed that partially inhibiting RNA polymerase I extended lifespan in the fruit fly. The manuscript by Sharifi takes this forward in that the authors further characterize the effects of reducing rRNA biogenesis to claim the longevity is due to metabolic changes. However, most of the findings are correlational and remain at the level of describing the phenotype of the long-lived animals. I think that direct demonstration that the longevity is due to metabolic changes would substantially strengthen the manuscript. This could be done in a way similar to examining the relationship between translation, rRNA biogenesis and longevity in the current manuscript (see e.g. Extended Data Figure 5).

We thank the reviewer for their kind assessment of our manuscript and their helpful suggestion of demonstrating the causal link between metabolic remodeling and longevity extension of Pol I inactivated animals. According to this recommendation and based on the hints obtained from proteomics and lipidomics analysis described on page 12, line 24 - page 13, line 20 and page 16, lines 6-16 of the revised manuscript, we decided to test the involvement of triglyceride preservation and mobilization (by knocking down the *atgl-1* gene) and mitochondrial biogenesis (by knock down of the *skn-1/Nrf2* gene) in the longevity benefits of Pol I inactivation. Interestingly, both knock downs abrogated life extension in nematodes exposed to RNAi against the core Pol I subunit *rpoa-2* (Fig. 5b and Fig. 6f), clearly demonstrating the key contribution of metabolic remodeling to geroprotective effects of Pol I inhibition.

Additionally, the authors should make clear note of the above-mentioned work in the introduction (the work is cited but not explained in sufficient detail) and include discussion of any differences in findings.

We have modified the revised manuscript to describe and discuss these previous findings in more detail and in the context of our novel contributions (page 2, lines 25 – page 3, line 5; page 6, lines 14-16; page 18, lines 4-8).

Furthermore:

For both ‘omics analyses, p values have to be adjusted to account for multiple comparisons.

Please make sure this is clear in the text. Details of the computational analyses performed should

be given, especially regarding significance testing and multiple testing corrections. I would expect the lipidomics dataset to also be deposited in a public database e.g. Metabolights as well.

The *p* values of proteomics analysis have been adjusted for multiple comparison by using the Bonferroni correction method. For lipidomics analysis, most suitable statistics methods were used in each specific case, and these are now described in the figure legends. The reference describing the computational analysis of the proteomics data has been added to the methods section to ensure clarity (methods file, page 7, line 3). The complete lipidomics data set has been deposited to the Metabolomics Workbench, and can be accessed at <http://dx.doi.org/10.21228/M84D89>. We thank the reviewer for helping us improve our presentation of the omics data.

Fig 2a and b – there should be controls for each transgenic construct alone as each insertion can have an effect i.e. tub-GAL80 and UAS controls are missing. Only 15 animals per condition? This does not sound correct. Additionally, please make explicit in the text that this lifespan was performed at high temperature. The transgenes do not appear to have been backcrossed to a common genetic background. They need to be.

We have included the tub-Gal80ts and UAS-IR control lines in the lifespan analyses and corrected the indication of sample sizes (for each lifespan ≥ 75 animals were used). Furthermore, the high temperature conditions used for lifespan analysis are now mentioned in the text (page 5, lines 9-10) and figure legend (Supplementary Fig. 1). As backcrossing of transgenes could not be achieved in the revision period, we have now performed the knockdown in two different UAS-IR lines ‘GD’ and ‘KK’ (Vienna Drosophila Recourse Center), to account for different genetic backgrounds. In both cases, knockdown of *Tif-IA* caused significant lifespan extension in comparison to each control line in both female and male flies (Supplementary Fig. 1d and e).

Page 4 lines 20-21, the elevated pre-rRNA in *ncl-1* kd is more likely to be due to accumulation of pre-rRNA due to an impairment in processing than due to increased pre-rRNA transcription, is it not?

NCL-1 inhibits expression of fibrillarin, thus *ncl-1* knockdown elevates fibrillarin levels. Thereby, pre-rRNA processing should be enhanced. Increased pre-rRNA synthesis is likely

resulting from an activating histone H2A methylation (H2AQ105me) at rDNA, which is mediated by fibrillarin (Tessarz et al., 2014). In response to this comment, we rephrased the text on page 5, lines 10-13 for better clarity.

Reviewer 4.

In this manuscript, the authors explored the effects of perturbation of pre-rRNA synthesis on longevity in *C. elegans* and *Drosophila* and identified that reduced pre-rRNA synthesis can improve lifespan through decrease of protein synthesis, preservation of energy and modulation of lipid metabolism. The data is in contrast to various studies in yeast and human cells have shown that rDNA instability and inhibition of Pol I activity can induce aging and senescence, respectively (Sharifi et al Annual review of Biochemistry, 2018; Hein et al the nucleolus and ribosome genes in aging and senescence, 2012).

Major concerns:

In Figure 4 d-e, the authors suggest that “Pol I inhibition by BMH-21 and CX-5461 compounds, promoted human cell survival under metabolic stress”. I find the data perplexing for the following reasons: 1) Inhibition of Pol I transcription has been shown to induce senescence in human fibroblasts (Quin et al 2016); 2) MTT assay may not be a reliable method for assessing cell survival if the effect of drug treatment is cell cycle arrest and senescence, 3) It seems that the combination effect of CX-5461/metformin and BMH-21/metformin is normalised to single agent CX-5461 and BMH-21 alone effects. If this is the case, data interpretation is misleading as it is likely masking the inhibitory effects of the Pol I inhibitors on cell proliferation. 4) Metformin has been shown to sensitise MYC-driven lymphoma cells to CX-5461/ everolimus therapy in vivo (Kusnadi et al., EMBO J 2020). The authors should discuss their findings in relation to previous research in various model systems that demonstrated the growth inhibitory effects of Pol I inhibition.

We thank the reviewer for these insightful comments. We have modified the respective figures and text to acknowledge the potential adverse effects of strong Pol I inhibition (page 11, lines 12-17), and have included new data to address the points 1-4 raised:

1) We have analyzed Pol I inhibition and senescence induction by CX-5461 and BMH-21. In accord with Quin et al (Quin et al., 2016), we found repression of pre-rRNA synthesis and p21

up-regulation in BJ cells (Supplementary Fig. 7a-d). Notably, both compounds were employed for demonstrating that pharmacological Pol I inhibition delivers comparable metabolic effects to the genetic ablation of Pol I activity, and we do not suggest that these stark compounds should be directly used as longevity drugs. This point is now clarified on page 11, lines 21-23.

2) We agree that MTT assay primarily measures mitochondrial activity, which is often used as a proxy read out of cell viability, and have rephrased the text accordingly (page 11, lines 15-18). Notably, CX-5461 and BMH-21 alleviate metformin-mediated mitochondrial impairment (Fig. 4e and f), supporting our findings in aging *C. elegans* (Fig. 4d).

3) The data were re-analyzed as recommended (Fig. 4e and f), showing dose-dependent but moderate effects of the Pol I inhibitors alone, while differences between metformin alone and metformin-inhibitor co-treated cells remained significant.

4) We thank the reviewer for bringing the link between CX-4561 and metformin in lymphoma cells to our attention. We included the discussion of these findings in the context of our data in the text page 11, line 23 – page 12, line 2.

In fact, the manuscript ignores a lot of work in mammalian model systems that show defects in Pol transcription leads to the induction of a nucleolar stress response and growth inhibitory phenotypes. Defects in ribosome biogenesis, including for example impaired pre-rRNA synthesis due to mutations in TCOF1, have been causatively linked to the pathogenesis of a group of diseases called ribosomopathies. Clinical manifestations of ribosomopathies include growth retardation and pre-mature aging (Hannan et al BBA 2013).

We agree that strong inhibition of pre-rRNA synthesis, especially early in development, has detrimental effects. In contrast, moderate Pol I inhibition that we achieve with the RNAi treatments in worms appears to be well tolerated, albeit the worms show transient growth retardation and are smaller than the control animals at young age (Fig. 2d). Moreover, ribosomopathies associated with premature aging, i.e. Werner and Blooms Syndrome, also entail genomic instability (Yamagata et al., 1998), which likely have a major impact on aging acceleration. In response to this comment, we now modified the text to discuss the potential adverse effects by untimely or excessive Pol I inhibition (page 19, line 16-25).

Additional comments:

In figure 2, the authors show reduced body size and neuromuscular performance in *tif-1A* deficient worms at day 2 of adulthood. What are the effects of *tif-1A* deficiency on the worm development at younger age? P53 activation plays a critical role in the pathogenesis of ribosomopathies due to activation of a nucleolar stress response. Does the knockdown of *tif-1A* affect p53 expression/activity in the model systems used in this study?

We found that rearing worms for 2 generations on *tif-1A* RNAi is required to achieve an effective knockdown. However, the worms are only slightly delayed in development, supporting the notion that moderate Pol I inhibition is well tolerable. While work by others (Wu et al., 2018a) has shown that nucleolar stress does indeed upregulate the *C. elegans* homolog of p53 (CEP-1), our mass spectrometry data did not detect increased levels of CEP-1 or its targets EGL-1 and CED-13 (Schumacher et al., 2005) in young or old worms deficient in *tif-1A* (Table S1).

Figure 1a-f showed that modulation of *tif-1A* expression level affected the lifespan. It is not clear at what stage (young, middle or old age) *tif-1A* expression was modulated and whether this effect is independent on the stage at which Pol I perturbation occurs.

As this information was only mentioned in the Material and Methods section, we have now included a sentence in the results part stating that the *tif-1A* knockdown was carried out for two generations (page 4, line 18-19). The strain carrying an additional *tif-1A* gene has a lifelong *tif-1A* overexpression. In Fig. 7, we use the highly efficient knockdown of the worm gene encoding the 2nd largest Pol I subunit (RPOA-2), and show that lifespan and healthspan can be improved by curbing pre-rRNA synthesis even late in adulthood.

The authors showed a significant downregulation of global protein synthesis in *tif-1A* deficient worms. Interestingly, *tif-1A* knockdown promoted a moderate but significant elevation of mitochondria levels at old age. This suggests that selective translational adaptation may underlay longevity under conditions of reduced rRNA synthesis. The authors can further analyze and identify proteins with upregulated expression levels in the proteomics data and the biological processes involved in this process.

The upregulation of mitochondrial ribosome observed in *tif-1A* inactivated worms is, in context of the co-detected enhanced metabolic stability (Fig. 4c and d), rather indicative of the preservation of mitochondrial content than of selective translational adaptations. In new experiments presented in Fig. 6c, d and f we tested the involvement of distinct mitochondrial quality control pathways in longevity extension of Pol I inactivated worms and found mitochondrial biogenesis to be essential (Fig. 6f), supporting our original interpretation of the mitochondrial ribosome proteomics.

Mitochondrial activity and function are associated with longevity and aging (Lima et al, Nature Aging 2, 199-213, 2022). While the authors showed *tif-1A* depletion preserved the mitochondrial levels and the ATP content in late life, there is no direct evidence to support maintenance of mitochondrial activity in *tif-1A* deficient worms. It is critical to measure mitochondrial oxidative phosphorylation by seahorse mitochondrial stress test to assess mitochondrial activity in *tif-1A* deficient worms.

In previous work we demonstrated that resilience to metformin toxicity is determined by enhanced mitochondrial function in both *C. elegans* and human fibroblasts, including the characterization of metformin responses by seahorse measurements (Espada et al., 2020). Here we used metformin resilience as a proxy method for demonstrating that Pol I inactivation indeed improves mitochondrial function in both experimental systems (Fig 4d, e and f).

Bibliography:

- Ermolaeva, M.A., Segref, A., Dakhovnik, A., Ou, H.L., Schneider, J.I., Utermohlen, O., Hoppe, T., and Schumacher, B. (2013). DNA damage in germ cells induces an innate immune response that triggers systemic stress resistance. *Nature* 501, 416-420.
- Espada, L., Dakhovnik, A., Chaudhari, P., Martirosyan, A., Miek, L., Poliezhaieva, T., Schaub, Y., Nair, A., Doring, N., Rahnis, N., *et al.* (2020). Loss of metabolic plasticity underlies metformin toxicity in aged *Caenorhabditis elegans*. *Nat Metab* 2, 1316-1331.
- Hansen, M., Taubert, S., Crawford, D., Libina, N., Lee, S.J., and Kenyon, C. (2007). Lifespan extension by conditions that inhibit translation in *Caenorhabditis elegans*. *Aging Cell* 6, 95-110.
- Holdorf, A.D., Higgins, D.P., Hart, A.C., Boag, P.R., Pazour, G.J., Walhout, A.J.M., and Walker, A.K. (2020). WormCat: An Online Tool for Annotation and Visualization of *Caenorhabditis elegans* Genome-Scale Data. *Genetics* 214, 279-294.
- Hsu, A.L., Murphy, C.T., and Kenyon, C. (2003). Regulation of aging and age-related disease by DAF-16 and heat-shock factor. *Science* 300, 1142-1145.

Murphy, C.T., McCarroll, S.A., Bargmann, C.I., Fraser, A., Kamath, R.S., Ahringer, J., Li, H., and Kenyon, C. (2003). Genes that act downstream of DAF-16 to influence the lifespan of *Caenorhabditis elegans*. *Nature* *424*, 277-283.

Mutlu, A.S., Duffy, J., and Wang, M.C. (2021). Lipid metabolism and lipid signals in aging and longevity. *Dev Cell* *56*, 1394-1407.

Palikaras, K., Lionaki, E., and Tavernarakis, N. (2015). Coupling mitogenesis and mitophagy for longevity. *Autophagy* *11*, 1428-1430.

Princz, A., Pelisch, F., and Tavernarakis, N. (2020). SUMO promotes longevity and maintains mitochondrial homeostasis during ageing in *Caenorhabditis elegans*. *Sci Rep* *10*, 15513.

Quin, J., Chan, K.T., Devlin, J.R., Cameron, D.P., Diesch, J., Cullinane, C., Ahern, J., Khot, A., Hein, N., George, A.J., *et al.* (2016). Inhibition of RNA polymerase I transcription initiation by CX-5461 activates non-canonical ATM/ATR signaling. *Oncotarget* *7*, 49800-49818.

Schumacher, B., Schertel, C., Wittenburg, N., Tuck, S., Mitani, S., Gartner, A., Conradt, B., and Shaham, S. (2005). *C. elegans* ced-13 can promote apoptosis and is induced in response to DNA damage. *Cell Death Differ* *12*, 153-161.

Tessarz, P., Santos-Rosa, H., Robson, S.C., Sylvestersen, K.B., Nelson, C.J., Nielsen, M.L., and Kouzarides, T. (2014). Glutamine methylation in histone H2A is an RNA-polymerase-I-dedicated modification. *Nature* *505*, 564-568.

Wu, J., Jiang, X., Li, Y., Zhu, T., Zhang, J., Zhang, Z., Zhang, L., Zhang, Y., Wang, Y., Zou, X., *et al.* (2018a). PHA-4/FoxA senses nucleolar stress to regulate lipid accumulation in *Caenorhabditis elegans*. *Nat Commun* *9*, 1195.

Wu, Z., Senchuk, M.M., Dues, D.J., Johnson, B.K., Cooper, J.F., Lew, L., Machiela, E., Schaar, C.E., DeJonge, H., Blackwell, T.K., *et al.* (2018b). Mitochondrial unfolded protein response transcription factor ATFS-1 promotes longevity in a long-lived mitochondrial mutant through activation of stress response pathways. *BMC Biol* *16*, 147.

Yamagata, K., Kato, J., Shimamoto, A., Goto, M., Furuichi, Y., and Ikeda, H. (1998). Bloom's and Werner's syndrome genes suppress hyperrecombination in yeast *sgs1* mutant: implication for genomic instability in human diseases. *Proc Natl Acad Sci U S A* *95*, 8733-8738.

REVIEWERS' COMMENTS:

Reviewer #1 (Remarks to the Author):

Authors addressed the reviewer's comments, and responded well by revising the manuscript. The responses and the revised manuscript have been reviewed as satisfactory. It seems a good shape for publication in Nature Communication.

Reviewer #2 (Remarks to the Author):

In this revised manuscript, the authors addressed the reviewer's major concern by demonstrating that reductions of Pol I activity and ribosome biogenesis may govern longevity via the same mechanism. Accordingly, they suggest that the major advance of the revised manuscript is now the underlying mechanism, which is "curtailment of Pol I activity reduces ATP expenditure, remodels the lipidome, and preserves mitochondrial function to promote longevity". However, this conclusion is not well supported by the current data. The majority of their research shows associations between lifespan and metabolic phenotypes, such as ATP, lipid metabolism, and mitochondrial homeostasis, but the mechanistic links between metabolic changes and longevity have not been conclusively established.

Major concerns:

1. As I mentioned in my previous review, it has not been established if ATP changes affect lifespan in response to reducing rRNA synthesis. Although the authors have revised the results section, the abstract also needs modification.
2. While the authors performed a comprehensive analysis of the lipidome and found very interesting correlations between lifespan and many specific lipid species, such as TAG, MUFA, PUFA, and PC/PE, the only mechanistic finding between the lipid metabolism and longevity is the *atgl-1* RNAi lifespan data, which is insufficient to support a crucial role of lipid metabolism in their model. Additional investigations demonstrating additional mechanistic links between lipids and lifespan are required to support their conclusion that "reduction of Pol I activity remodels the lipidome to increase longevity." For example, which lipid alterations, such as PUFA and PC/PE, as implied by the authors, affect lifespan?
3. The authors provided more data on mitochondrial changes, which are very important. But they only used *skn-1* mutants to demonstrate a mechanistic link between mitochondria and

lifespan, which I think is not appropriate. SKN-1 is a transcript factor regulating numerous cellular processes, such as detoxification, lipid metabolism, immune response, and mitochondria. Thus, SKN-1 involvement is not always linked to mitochondrial biogenesis. To build a link between mitochondria and lifespan, other mitochondrial factors, such as fusion and fission genes and mitochondrial intrinsic biogenesis genes, need to be examined.

4. As the mechanism is now the major advance of the revised paper, it is important to demonstrate the relationship between lipid metabolism and mitochondrial phenotypes. Whether lipid alterations work upstream of mitochondrial homeostasis or in the reverse direction.

Minor concerns:

1. As the authors stated, SKN-1 has been reported to respond to lipid changes. So whether it is the case in their model?
2. The use of prx-5 RNAi to demonstrate the involvement of peroxisomal fatty acid oxidation is insufficient. Since they focused on mitochondria, how about the role of mitochondrial fatty acid oxidation?

Reviewer #3 (Remarks to the Author):

The authors have addressed the majority of my previous comments. However, there is one major omission: no backcrossing for fruit fly lifespans.

Due to susceptibility of *Drosophila* lifespan to inbreeding depression, performing lifespans in this species without correct backcrossing of the parental lines will result in systematic false positives (an extension of lifespan due to hybrid vigour). This cannot be controlled for by simply using yet another, non-backcrossed RNAi line. Hence, as the genetic background was not standardised (with backcrossing) for the *Drosophila* lifespan experiments, in addition to the high temperature resulting in very short survival times, I think these experiments were overall too poorly performed to be published. As these experiments are not essential to the manuscript, I would request they be removed before the publication of the manuscript.

Reviewer #4 (Remarks to the Author):

While the manuscript is improved, the observations remain correlative. There is no direct mechanism to support that modulation of rRNA gene activity directly impacts mitochondrial homeostasis and lipid metabolism and influences fitness and longevity. Modulation of Pol I activity influences translation and other cellular processes including cell cycle progression and proliferation and this in turn could impact mitochondrial homeostasis and cellular energy.

Major concerns:

The data in Fig 7a-d is contradictory to the conclusions in the paper. Despite no change in pre-rRNA levels in AD6 and AD8 in *rpoa-2* RNAi samples, a prolonged lifespan was detected. An explanation is required.

The authors should expand on the implication that *tif-1A* mRNA and pre-rRNA levels were efficiently diminished only when worms were reared for two generations on double-stranded RNA (dsRNA) producing bacteria (Fig. 1d, e). Why does KD of *rpoa-2* happens quicker and not *Tif1-a*? Is this expected in this model using this RNAi approach? This is concerning because in many experiments the two KDs are used interchangeable as a model of Pol perturbation however the dynamics of KD are different and this can influence the outcome.

The paper suggests that “the curtailment of pre-rRNA synthesis attenuates metabolic aging and stabilizes energy turnover, providing an ATP surplus for healthy homeostasis.” How can ATP surplus rescue perturbations in translation and a nucleolar stress response to promote longevity? The data presented here is focused on metabolic pathways and the study ignores other cellular and molecular pathways that can be impacted by perturbing Pol I activity. The conclusions should be toned down to focus on the link between metabolism and life span.

I also find the data in Fig 5 confusing. Figure 5A should include *rpoa-2* RNAi. The authors should also show the effect of *atgl-1* RNAi and the combined KD with *rpoa-2* on TAGs levels. The conclusions are overstating the results without proper controls.

Other comments:

Fig 6b should include tif-1A KD.

To improve clarity, when referring to modulating Pol I activity the authors should use “reducing Pol I transcription” and not “inhibition of Pol I transcription”.

Typo on line 14, page 11 "senesce" should be senescence.

We thank all reviewers for their efforts in assessing our revised manuscript. To fully address the remaining open questions, we revised the previous manuscript version by adding new experiments and amending the text accordingly (highlighted in grey). Please find our responses to the specific comments below.

Reviewer #1 (Remarks to the Author):

Comment: *Authors addressed the reviewer's comments, and responded well by revising the manuscript. The responses and the revised manuscript have been reviewed as satisfactory. It seems a good shape for publication in Nature Communication.*

Response: We thank the reviewer for their positive assessment of the revised manuscript.

Reviewer #2 (Remarks to the Author):

Comment: *In this revised manuscript, the authors addressed the reviewer's major concern by demonstrating that reductions of Pol I activity and ribosome biogenesis may govern longevity via the same mechanism. Accordingly, they suggest that the major advance of the revised manuscript is now the underlying mechanism, which is "curtailment of Pol I activity reduces ATP expenditure, remodels the lipidome, and preserves mitochondrial function to promote longevity". However, this conclusion is not well supported by the current data. The majority of their research shows associations between lifespan and metabolic phenotypes, such as ATP, lipid metabolism, and mitochondrial homeostasis, but the mechanistic links between metabolic changes and longevity have not been conclusively established.*

Response: In our newly revised manuscript, we acted on the reviewer's helpful suggestions and further strengthened the causal and functional links between Pol I activity, lipid and mitochondrial homeostasis, and longevity. Our new conclusive findings and the updated mechanistic model are laid out below and in the newly revised sections of the manuscript (page 14, line 5 – page 18, line 20; page 20, line 22 – page 22, line 11).

Major concerns:

1. *As I mentioned in my previous review, it has not been established if ATP changes affect lifespan in response to reducing rRNA synthesis. Although the authors have revised the results section, the abstract also needs modification.*

Response: In response to this comment, we revised the abstract text, placing focus on lipidome remodeling and mitochondrial effects of Pol I downregulation (page 1, line 22 - page 2, line 4). The amended statements are supported by the new data provided.

2. *While the authors performed a comprehensive analysis of the lipidome and found very interesting correlations between lifespan and many specific lipid species, such as TAG, MUFA, PUFA, and PC/PE, the only mechanistic finding between the lipid metabolism and longevity is the *atgl-1* RNAi lifespan data, which is insufficient to support a crucial role of lipid metabolism in their model. Additional investigations demonstrating additional mechanistic links between lipids and lifespan are required to support their conclusion that "reduction of Pol I activity remodels the lipidome to increase longevity." For example, which lipid alterations, such as PUFA and PC/PE, as implied by the authors, affect lifespan?*

Response: In response to this comment, we reassessed our proteomics data to search for mediators of lipid metabolism that are (a) known to influence longevity, (b) are differentially expressed between control and *tif-1A* KD animals during aging and (c) whose differential expression matches the lipidome profiles of the respective backgrounds. The combination of these three criteria revealed two main candidates – fatty acid CoA synthetase ACS-2 implicated in the longevity effects of fasting and metformin (Pryor et al., 2019; Van Gilst et al., 2005) and $\Delta 9$ FA desaturase FAT-5 implicated in longevity promotion by increased abundance of MUFA lipids (Han et al., 2017; Xiao et al., 2023) (of note, other $\Delta 9$ FA desaturases such as *fat-6* and *fat-7* were not comparably regulated in our data set). Although both candidates were up-regulated in worms with reduced Pol I activity (Supplementary Fig. 14e), and their respective RNAi interventions were effective (Supplementary Fig. 14c, d), the knockdown of *acs-2* or *fat-5* did not abrogate the life-extension induced by RNAi against Pol I subunit *rpoa-2* (Fig. 6d and Supplementary Fig. 14b). Thus, we conclude that at least some of the lipidome changes induced by perturbation of Pol I activity occur without having an active impact on longevity, as now stated in the revised manuscript (page 17, lines 5-15).

Having ACS-2 and FAT-5 ruled out, we focused on the TAG lipase ATGL-1 because TAG levels at middle and old age were among the strongest differences between lipidomes of Pol I down- (*tif-1A KD*) and up-regulated (*ncl-1 KD*) worms, respectively (Fig. 5a). Based on our data and a recent report connecting accumulation of TAGs in lipid droplets with extended longevity and a youthful lipidome profile in nematodes exposed to elevated levels of MUFA lipids (Papsdorf et al., 2023), we considered differential TAG turnover as the possible central mechanism connecting modulation of Pol I activity to the respective metabolic and lifespan changes. This hypothesis is in line with our previous observation that knockdown of *atgl-1* abrogates longevity benefits of the *rpoa-2* RNAi treated animals (current Fig. 6a), and we now further tested it by combining *atgl-1* and *ncl-1* knockdowns. Strikingly, ATGL-1 depletion rescued the shortened lifespan of *ncl-1* inactivated worms (Fig. 6b) linking differential TAG lipolysis to both lifespan extension and shortening downstream of reduced and enhanced Pol I activity respectively. These findings fit to the established role of ATGL-1 as a functional sensor of energy supply: ATGL-1 is known to be activated during starvation to facilitate energy extraction from TAGs (Lee et al., 2014), a response which could also be triggered by the high energy demand of elevated ribosome biogenesis in *ncl-1* KD worms. Conversely, ATGL-1 activity is dampened in *C. elegans* dauer larvae supporting their exceptional longevity by rationing TAG consumption over longer time (Narbonne and Roy, 2009; Xie and Roy, 2015), a scenario reminiscent of the attenuated energy expenditure in *tif-1A* and *rpoa-2* KD animals (models in Fig. 6f and Supplementary Fig. 16).

3. *The authors provided more data on mitochondrial changes, which are very important. But they only used *skn-1* mutants to demonstrate a mechanistic link between mitochondria and lifespan, which I think is not appropriate. SKN-1 is a transcript factor regulating numerous cellular processes, such as detoxification, lipid metabolism, immune response, and mitochondria. Thus, SKN-1 involvement is not always linked to mitochondrial biogenesis. To build a link between mitochondria and lifespan, other mitochondrial factors, such as fusion and fission genes and mitochondrial intrinsic biogenesis genes, need to be examined.*

Response: We agree that the effects of SKN-1 are pleiotropic and include roles besides mitochondrial biogenesis, and we have now rephrased the manuscript text to describe this clearly

(page 15, lines 18-25). To further validate the connection between Pol I activity and mitochondria, we followed the suggestion by the referee and conducted Pol I perturbation in mutants lacking mitochondrial fission and fusion, which conferred lifespan extension to a similar extent as in wild-type worms (Supplementary Fig. 12c and d). Thus, fission and fusion, as well as the previously tested mitochondrial UPR and mitophagy (current Fig. 5e and Supplementary Fig. 13a) appeared not to be implicated in Pol I-dependent longevity. To further investigate the connection between rDNA transcription and mitochondrial activity, we combined *rpoa-2* RNAi with dose-dependent poisoning of mitochondria by the uncoupling agent FCCP (Brennan et al., 2006). We found the full longevity benefit of *rpoa-2* KD to be maintained only at lowest concentration of the drug, while higher FCCP doses diminished and even reversed the life-extension of RPOA-2 depleted animals (Fig. 5f and Supplementary Fig. 13b, c). This result confirmed functional importance of mitochondria for Pol I-linked longevity and demonstrated the ability of curbed rDNA transcription to buffer moderate mitochondrial stress. The specific mechanism that links reduced Pol I activity, mitochondrial stress and longevity is addressed in the response to the comment below.

4. *As the mechanism is now the major advance of the revised paper, it is important to demonstrate the relationship between lipid metabolism and mitochondrial phenotypes. Whether lipid alterations work upstream of mitochondrial homeostasis or in the reverse direction.*

Response: We thank the referee for this query that was very helpful for the assembly of our current model, which shows that lipid alterations are upstream of mitochondrial homeostasis. We investigated this hierarchy by assessing mitochondrial stress upon manipulation of TAG turnover by *atgl-1* RNAi in *tif-1A* and *ncl-1* KD backgrounds. Using the *hsp-6p::GFP* reporter as a readout, we found that high mitochondrial stress levels of *ncl-1* KDs were dampened by *atgl-1* RNAi, while no change was seen in *tif-1A* KD worms already showing very low baseline reporter expression (Supplementary Fig. 15a, b). This difference, along with distinct TAG turnover patterns of the 2 KDs seen by lipidomics, suggested that ATGL-1 activity may be intrinsically increased in Pol-I boosted animals and lowered in Pol I-inactivated ones. The finding also points to the causal role of differential TAG lipolysis in both mitochondrial protection and mitochondrial impairment upon Pol I modulation. Interestingly, a recent study

connected increased lipolysis of TAGs and the subsequently boosted flux of fatty acids into mitochondria to alterations in mitochondrial homeostasis and elevated oxygen consumption rate (OCR) (Sharma et al., 2023). We therefore tested oxygen consumption by whole worm Seahorse analysis in *tif-1A* and *ncl-1* KD animals on adulthood day 12, when strongest differences in TAG expenditure were seen between these KDs (Fig. 5a) and found OCRs to be lowered and elevated, respectively (Fig. 6e). These data are in line with the model that elevated energy demand of ribosome biogenesis (as in *ncl-1* KD) leads to elevated TAG turnover, which subsequently increases OCR and mitochondrial stress, finally contributing to premature aging. In contrast, reduced energy expenditure by dampened Pol I activity curbs TAG catabolism, lowers mitochondrial stress and thereby attenuates metabolic aging (Fig 6f. and Supplementary Fig. 16). In this context, we observed that *skn-1* mutants not only fail to benefit from the geroprotective effects of Pol I inhibition (current Fig. 5g), but also they fail to accumulate TAGs in the context of *rpoa-2* KD contrary to wild type controls (Supplementary Fig. S14a). This finding suggests that SKN-1 might be implicated in the regulation of TAG lipolysis downstream of rDNA transcription, which would match its previously reported role in regulating lipid expenditure (Steinbaugh et al., 2015), but further follow up studies are required to systematically validate this hypothesis.

Minor concerns:

1. *As the authors stated, SKN-1 has been reported to respond to lipid changes. So whether it is the case in their model?*

Response: As discussed in the responses to the comments above, our data on the links between SKN-1 and lipid changes triggered by modulated Pol I activity is limited. Specifically, we do not have experimental evidence of SKN-1 responding to lipid changes. We therefore revised the respective parts of the manuscript for clarity (page 15, lines 18 – 25; page 18, lines 14-20).

2. *The use of prx-5 RNAi to demonstrate the involvement of peroxisomal fatty acid oxidation is insufficient. Since they focused on mitochondria, how about the role of mitochondrial fatty acid oxidation?*

Response: In our study, we used loss of function mutant of the peroxisome assembly factor *prx-5* (*prx-5(ku517)* allele) to test the involvement of peroxisomes in Pol I-dependent longevity. Inactivation of *prx-5* is well known to impair peroxisome activity (Rackles et al., 2021) and is frequently used to test if peroxisomal functions, such as β -oxidation of very long chain FAs, are involved in specific phenotypes (Pryor et al., 2019). For example, a recent study linking elevated MUFA content to extended longevity in nematodes utilized *prx-5* RNAi to demonstrate the key role of peroxisomes in this process (Papsdorf et al., 2023). To improve clarity, we re-phrased the manuscript text with specific reference to whole organelle function being perturbed rather than only β -oxidation (page 17, lines 3-5).

At the same time, we followed the reviewer's suggestion and tested mitochondrial FA β -oxidation in the context of Pol I modulation by using RNAi against a crucial mediator of mitochondrial FA β -oxidation *acs-2* (Ramachandran et al., 2019). As indicated above, depletion of ACS-2, did not diminish longevity benefits of *rpoa-2* RNAi, despite clear upregulation of the ACS-2 protein in Pol I inactivated worms (Fig. 6d, Supplementary Fig. 14c and 14e). Together with our experiments involving *atgl-1*, *prx-5* and *fat-5*, this finding highlights the specific reliance of Pol I longevity on differential TAG catabolism.

We would like to thank the referee for their thorough comments, which allowed us to assemble and present a more comprehensive and detailed mechanistic model.

Reviewer #3 (Remarks to the Author):

The authors have addressed the majority of my previous comments.

Response: We thank the referee for this positive assessment.

However, there is one major omission: no backcrossing for fruit fly lifespans.

Due to susceptibility of Drosophila lifespan to inbreeding depression, performing lifespans in this species without correct backcrossing of the parental lines will result in systematic false positives (an extension of lifespan due hybrid vigour). This cannot be controlled for by simply using yet another, non-backcrossed RNAi line. Hence, as the genetic background was not

standardised (with backcrossing) for the Drosophila lifespan experiments, in addition to the high temperature resulting in very short survival times, I think these experiments were overall too poorly performed to be published. As these experiments are not essential to the manuscript, I would request they be removed before the publication of the manuscript.

Response: Unfortunately, we misunderstood the meaning of the reviewer's initial query as a question of whether longevity benefits of Tif-1A inactivation are background specific, e.g. only detectable in the specific RNAi line presented. This misinterpretation guided our response of providing additional non-backcrossed RNAi lines. We now understand the reviewer's point of view about the possibility of hybrid vigor and fully agree with the necessity of backcrossing in this context. However, we cannot perform backcrossing due to time constraints. We thus complied with the reviewer's request to omit the data.

Reviewer #4 (Remarks to the Author):

While the manuscript is improved, the observations remain correlative. There is no direct mechanism to support that modulation of rRNA gene activity directly impacts mitochondrial homeostasis and lipid metabolism and influences fitness and longevity.

Response: We thank the reviewer for acknowledging the advancement of our revised manuscript. Given that rRNA gene transcript affects a multitude of cellular mechanisms, we agree that a direct link to different mechanisms is challenging to prove. However, our previous and newly added data show unambiguously that Pol I activity impinges on TAG metabolism, which in turn determines mitochondrial OCR and stress levels. This novel link between rDNA transcription levels and mitochondria has a key impact on lifespan and health span of nematodes. The supporting new evidence is presented in revised Figures 5 and 6, and in revised Supplementary Figures 12-15, the updated mechanistic model is shown in current Fig 6f and Supplementary Fig. 16, and the updated concept is described on page 14, line 5 – page 18, line 20 as well as in page 20, line 22 – page 22, line 11 of the current manuscript. We hope, that with these updates we can convince the reviewer of the causal link between Pol I activity, lipid metabolism and mitochondria in the regulation of longevity.

Modulation of Pol I activity influences translation and other cellular processes including cell cycle progression and proliferation and this in turn could impact mitochondrial homeostasis and cellular energy.

Response: All of our longevity experiments were conducted in adult *C. elegans* hermaphrodites harboring exclusively post-mitotic somatic organs (Olsen et al., 2006). This feature makes *C. elegans* an ideal model for investigating somatic cellular maintenance during aging and concurrently precludes indirect contribution of cell cycle and proliferation. The impact of ribosome biogenesis and translation on the longevity effects of reduced Pol I activity has been extensively dissected already in the first two versions of the manuscript (current Fig. 3a-d, ribosome data set; Fig. 4a-b and Supplementary Fig. S6a-c). Our findings demonstrated that reduced rRNA gene transcription affects protein synthesis (Supplementary Fig 6a), but this is not the only driving mechanism of life extension in this case (Fig 4a and Supplementary Fig. 6b). Notably, curbed translation does not suffice for the energy saving comparable to the effect of reduced Pol I activity, as directly tested in Fig. 4b. We hope to have convinced the reviewer by explicitly highlighting the relevant data in our current response.

Major concerns:

*The data in Fig 7a-d is contradictory to the conclusions in the paper. Despite no change in pre-rRNA levels in AD6 and AD8 in *rpoa-2* RNAi samples, a prolonged lifespan was detected. An explanation is required.*

Response: We thank the reviewer for pointing out the need for an additional clarification here. The simple explanation is that the lifespan analysis and testing of pre-rRNA and *rpoa-2* mRNA levels have different timelines. The measurements of pre-rRNA levels (Fig. 7c) and *rpoa-2* RNAi efficiency (Fig. 7b) were performed on day 10 of adulthood to demonstrate that time is needed for the RNAi to take effect in old animals (the strongest changes in expression are seen after 10 days of *rpoa-2* RNAi exposure in both cases, while trends of changes are observed after 4 and 2 days of RNAi feeding). At the same time, lifespan analyses were conducted for up to 30 days (Fig. 7d). Thus, the *rpoa-2* RNAi had for all conditions outlined in Fig 7a sufficient time to induce the effective knockdown and thus prolong lifespan (Fig. 7d). The experiment was set up

in this specific manner to elucidate the longevity effects of very late in life depletion of Pol I activity. We have amended the text (page 19, lines 2-11) to clarify this circumstance.

*The authors should expand on the implication that *tif-1A* mRNA and pre-rRNA levels were efficiently diminished only when worms were reared for two generations on double-stranded RNA (dsRNA) producing bacteria (Fig. 1d, e). Why does KD of *rpoa-2* happens quicker and not *tif-1A*? Is this expected in this model using this RNAi approach? This is concerning because in many experiments the two KDs are used interchangeable as a model of Pol I perturbation however the dynamics of KD are different and this can influence the outcome.*

Response: It is very common in the *C. elegans* model that RNAi treatments differ in their knockdown efficiencies, depending on the function and expression levels of the target genes. Consequently, some targets (like *tif-1A* in our case) require inactivation from the earliest developmental stage, which is achieved by RNAi-exposure of hermaphrodite mothers that carry developing embryos (Shiu and Hunter, 2017). Notably, this parental RNAi exposure only facilitates the gene inactivation process in the progeny (our test animals) and does not interfere with the dynamics of the gene knockdown, allowing us to use *tif-1A* RNAi and the faster acting *rpoa-2* RNAi as equally valid tools for reducing Pol I activity.

The paper suggests that “the curtailment of pre-rRNA synthesis attenuates metabolic aging and stabilizes energy turnover, providing an ATP surplus for healthy homeostasis.” How can ATP surplus rescue perturbations in translation and a nucleolar stress response to promote longevity? The data presented here is focused on metabolic pathways and the study ignores other cellular and molecular pathways that can be impacted by perturbing Pol I activity. The conclusions should be toned down to focus on the link between metabolism and life span.

Response: We respectfully disagree with this comment, and we are confident to have explored all reasonable mechanisms. We certainly did not neglect the possible induction of stress and stress responses by the moderate inactivation of Pol I. For example, in Fig. 3a-b, we explicitly tested the modulation of the key stress responses by *tif-1A* and *ncl-1* KDs, observing no obvious up-regulation upon reduced Pol I activity. Thus, we conclude that the moderate perturbation of

Pol I that confers longevity benefits, does not cause severe stress, which is possibly one reason behind its life extension properties. We discuss the lack of stress responses under our chosen Pol I inactivation regimen in the discussion section of the current manuscript on **page 20, lines 17-22**.

On the other hand, our unbiased proteomics analysis clearly revealed the terms ribosome and metabolism as the two most strongly affected categories by both *tif-1A* and *ncl-1* RNAi treatments (**Fig. 3c, d**). This finding prompted us to address the functional involvement of both translation and metabolism in Pol I-dependent longevity, discovering the unexpectedly prevailing role of mitochondria and lipid metabolism. We thus believe to have extensively explored all relevant mechanistic possibilities by both candidate-based and unbiased experiments, and we hope the reviewer would agree with our point of view following this detailed explanation.

*I also find the data in Fig 5 confusing. Figure 5A should include *rpoa-2* RNAi. The authors should also show the effect of *atgl-1* RNAi and the combined KD with *rpoa-2* on TAGs levels. The conclusions are overstating the results without proper controls.*

Response: Given that both TIF-1A and RPOA-2 are part of the Pol I transcription machinery, the depletion of either factor should affect initiation of rRNA gene transcription. Accordingly, our *tif-1A* and *rpoa-2* RNAi experiments show a comparable reduction in pre-rRNA levels and lifespan extension (**Fig. 1 d-i**). While *tif-1A* and *rpoa-2* KDs are used simultaneously in certain key experiments, it is not feasible to do so in more elaborate analyses, like lipidomics. However, based on the reviewer's concern regarding Fig. 5a, we have now validated the increased TAG levels in *rpoa-2* KD animals with an alternative approach – the Oil Red O whole animal lipid staining, which predominantly stains TAGs (Ramirez-Zacarias et al., 1992) (**Supplementary Fig. 14a**).

Other comments:

*Fig 6b should include *tif-1A* KD.*

Response: As argued above, we are confident, based on extensive prior knowledge in the field and our own data, that *tif-1A* and *rpoa-2* RNAi treatments induce the same molecular and cellular responses. Therefore, we have used in many new experiments the faster acting *rpoa-2* RNAi to avoid time delays. The particular experiment shown in former Fig 6b (**current Supplementary Fig. 12b**) is complementary to lifespan tests presented in **Fig. 5e, Supplementary Fig. S12 c, d and Supplementary Fig. S13a**. All of these tests were conducted by using *rpoa-2* inactivation. It was therefore reasonable to use *rpoa-2* KD also in this experiment, while addition of *tif-1A* RNAi would not advance the conceptual value of the result.

To improve clarity, when referring to modulating Pol I activity the authors should use “reducing Pol I transcription” and not “inhibition of Pol I transcription”.

Response: Appropriate changes were introduced throughout the text (purple marked text).

Typo on line 14, page 11 "senesce" should be senescence.

Response: We thank the reviewer for spotting this typo, it has been corrected.

Bibliography:

- Brennan, J.P., Berry, R.G., Baghai, M., Duchen, M.R., and Shattock, M.J. (2006). FCCP is cardioprotective at concentrations that cause mitochondrial oxidation without detectable depolarisation. *Cardiovasc Res* 72, 322-330.
- Han, S., Schroeder, E.A., Silva-Garcia, C.G., Hebestreit, K., Mair, W.B., and Brunet, A. (2017). Mono-unsaturated fatty acids link H3K4me3 modifiers to *C. elegans* lifespan. *Nature* 544, 185-190.
- Lee, J.H., Kong, J., Jang, J.Y., Han, J.S., Ji, Y., Lee, J., and Kim, J.B. (2014). Lipid droplet protein LID-1 mediates ATGL-1-dependent lipolysis during fasting in *Caenorhabditis elegans*. *Mol Cell Biol* 34, 4165-4176.
- Narbonne, P., and Roy, R. (2009). *Caenorhabditis elegans* dauers need LKB1/AMPK to ration lipid reserves and ensure long-term survival. *Nature* 457, 210-214.
- Olsen, A., Vantipalli, M.C., and Lithgow, G.J. (2006). Checkpoint proteins control survival of the postmitotic cells in *Caenorhabditis elegans*. *Science* 312, 1381-1385.
- Papsdorf, K., Miklas, J.W., Hosseini, A., Cabruja, M., Morrow, C.S., Savini, M., Yu, Y., Silva-Garcia, C.G., Haseley, N.R., Murphy, L.M., *et al.* (2023). Lipid droplets and peroxisomes are co-regulated to drive lifespan extension in response to mono-unsaturated fatty acids. *Nat Cell Biol* 25, 672-684.

Pryor, R., Norvaisas, P., Marinos, G., Best, L., Thingholm, L.B., Quintaneiro, L.M., De Haes, W., Esser, D., Waschina, S., Lujan, C., *et al.* (2019). Host-Microbe-Drug-Nutrient Screen Identifies Bacterial Effectors of Metformin Therapy. *Cell* *178*, 1299-1312 e1229.

Rackles, E., Witting, M., Forne, I., Zhang, X., Zacherl, J., Schrott, S., Fischer, C., Ewbank, J.J., Osman, C., Imhof, A., *et al.* (2021). Reduced peroxisomal import triggers peroxisomal retrograde signaling. *Cell Rep* *34*, 108653.

Ramachandran, P.V., Savini, M., Folick, A.K., Hu, K., Masand, R., Graham, B.H., and Wang, M.C. (2019). Lysosomal Signaling Promotes Longevity by Adjusting Mitochondrial Activity. *Dev Cell* *48*, 685-696 e685.

Ramirez-Zacarias, J.L., Castro-Munozledo, F., and Kuri-Harcuch, W. (1992). Quantitation of adipose conversion and triglycerides by staining intracytoplasmic lipids with Oil red O. *Histochemistry* *97*, 493-497.

Sharma, A.K., Wang, T., Othman, A., Khandelwal, R., Balaz, M., Modica, S., Zamboni, N., and Wolfrum, C. (2023). Basal re-esterification finetunes mitochondrial fatty acid utilization. *Mol Metab* *71*, 101701.

Shiu, P.K., and Hunter, C.P. (2017). Early Developmental Exposure to dsRNA Is Critical for Initiating Efficient Nuclear RNAi in *C. elegans*. *Cell Rep* *18*, 2969-2978.

Steinbaugh, M.J., Narasimhan, S.D., Robida-Stubbs, S., Moronetti Mazzeo, L.E., Dreyfuss, J.M., Hourihan, J.M., Raghavan, P., Operana, T.N., Esmailie, R., and Blackwell, T.K. (2015). Lipid-mediated regulation of SKN-1/Nrf in response to germ cell absence. *Elife* *4*.

Van Gilst, M.R., Hadjivassiliou, H., and Yamamoto, K.R. (2005). A *Caenorhabditis elegans* nutrient response system partially dependent on nuclear receptor NHR-49. *Proc Natl Acad Sci U S A* *102*, 13496-13501.

Xiao, Y., Liu, F., Zhu, X., Li, S., Meng, L., Jiang, N., Yu, C., Wang, H., Qin, Y., Hui, J., *et al.* (2023). Dioscin integrates regulation of monosaturated fatty acid metabolism to extend the life span through XBP-1/SBP-1 dependent manner. *iScience* *26*, 106265.

Xie, M., and Roy, R. (2015). AMP-Activated Kinase Regulates Lipid Droplet Localization and Stability of Adipose Triglyceride Lipase in *C. elegans* Dauer Larvae. *PLoS One* *10*, e0130480.

REVIEWERS' COMMENTS

Reviewer #2 (Remarks to the Author):

The authors did an excellent job in addressing the reviewer's concerns. The novelty of their findings is clearly presented in the current working model. It is a strong candidate for publication in Nature Communications.

Reviewer #5 (Remarks to the Author):

This is an interesting manuscript, raising the exciting hypothesis that lifespan effects of Pol-I inhibition may not only act via changes in translation, but also via changing the cellular energy homeostasis. Clearly separating the causal contributions of each aspect is difficult, which has led to multiple revisions and extensive new data. Overall, the comments of Reviewer 4 have been sufficiently addressed. Here are some remaining comments, mostly regarding the wording of conclusions.

1. rRNA inhibition vs translation inhibition (longevity source):

"We, therefore, concluded that perturbation of pre-rRNA synthesis likely confers additional pro-longevity effects beyond curbing protein synthesis."; "...surpass those of translational inhibition"

On page 9 paragraph line 5 the authors conclude that most of the contribution to lifespan extension due to RNA Pol-I inhibition (tif-1A/rpoa-2 RNAi; 27.3%/33.3%) comes from effects upstream of reduced translation, like lower rRNA production or consequent lower Ribosome assembly, instead of reduced translation.

They justify this by affecting the axis (Pol-I -> rRNA -> ribo -> translation -> lifespan) more downstream by inhibiting specifically translation through ife-2 RNAi (translation initiation factor) or CHX (elongation termination) and showing it barely increases lifespan (9.5% or 10% respectively) compared to the tif-1A/rpoa-2 RNAi increase.

Without proving that ife-2 RNAi or CHX are lowering translation to similar levels of tif-1A/rpoa-2 RNAi, these observations are not enough to prove RNA Pol-I inhibition-related lifespan increase is significantly not due to reduced translation.

The same applies to the interactions of tif-1A RNAi with ife-2 RNAi and tif-1A RNAi with CHX.

It is possible that there is no further lifespan increase, because the *ife-2* RNAi and CHX translation reduction is much smaller than the one caused by *tif-1A/rpoa-2* RNAi and becomes dominated by the latter. Ideally one should provide evidence similar to Figure S6.a to clarify and assess the extent of translation inhibition also in *ife-2* RNAi, CHX and respective interactions. However, this manuscript already went through multiple revisions. In order to not further delay the publication, one may adopt the wording accordingly and discuss the limitations in the Discussion section.

On page 10 line 3 "indicating that direct translational repression does not preserve cellular energy as efficiently as inhibition of pre-rRNA synthesis, which in turn affects all subsequent steps of ribosomal biogenesis". For that we would need to know if translation is being equally repressed.

2. rRNA inhibition sustains mitochondrial function?

The authors conclude from the metformin experiments (page 10/11): "This result indicated that restriction of rRNA 1 synthesis sustains mitochondrial function and metabolic plasticity during aging in *C. elegans*."

I don't think the specific metformin experiments show that mitochondrial function is better maintained. I think they only confirm the increased metabolic plasticity. I suggest to word the sentence as follows:

"This result indicated that restriction of rRNA 1 synthesis sustains energy metabolism and metabolic plasticity during aging in *C. elegans*."

Minor points

3. Figure 5a: label on the x-axis is missing.

4. Figure S4/S5 has not only an unnecessary but also meaningless x-axis if you just call it "bubbles_z" without further explanation.

Reviewer #6 (Remarks to the Author):

The authors addressed the majority of the comments of this Reviewer in the previously revised manuscript. However, the Reviewer is still concerned about the fly *Drosophila* experiments and indicated some important controls (via standardized fly backcrossing) were missing to substantiate the lifespan fly data. The Reviewer states in the current form the fly data are therefore not solid enough for publication and suggested to remove them since they are not essential to the manuscript. The authors indicated there has been a misunderstanding on the original Reviewers request and agreed to remove the fly data from the manuscript.

We thank the reviewers for their time and effort in assessing our revised manuscript, as well as for their final feedback. The specific responses to reviewer's comments are included below.

Reviewer #2 (Remarks to the Author):

The authors did an excellent job in addressing the reviewer's concerns. The novelty of their findings is clearly presented in the current working model. It is a strong candidate for publication in Nature Communications.

Response: We thank the reviewer for their previous helpful suggestions and for their positive final assessment.

Reviewer #5 (Remarks to the Author):

This is an interesting manuscript, raising the exciting hypothesis that lifespan effects of Pol-I inhibition may not only act via changes in translation, but also via changing the cellular energy homeostasis. Clearly separating the causal contributions of each aspect is difficult, which has led to multiple revisions and extensive new data. Overall, the comments of Reviewer 4 have been sufficiently addressed. Here are some remaining comments, mostly regarding the wording of conclusions.

1. rRNA inhibition vs translation inhibition (longevity source):

"We, therefore, concluded that perturbation of pre-rRNA synthesis likely confers additional pro-longevity effects beyond curbing protein synthesis."; "...surpass those of translational inhibition"

On page 9 paragraph line 5 the authors conclude that most of the contribution to lifespan extension due to RNA Pol-I inhibition (*tif-1A/rpoa-2* RNAi; 27.3%/33.3%) comes from effects upstream of reduced translation, like lower rRNA production or consequent lower ribosome assembly, instead of reduced translation.

They justify this by affecting the axis (Pol-I -> rRNA -> ribo -> translation -> lifespan) more downstream by inhibiting specifically translation through *ife-2* RNAi (translation initiation factor) or CHX (elongation termination) and showing it barely increases lifespan (9.5% or 10% respectively) compared to the *tif-1A/rpoa-2* RNAi increase.

Without proving that *ife-2* RNAi or CHX are lowering translation to similar levels of *tif-1A/rpoa-2* RNAi, these observations are not enough to prove RNA Pol-I inhibition-related lifespan increase is significantly not due to reduced translation.

The same applies to the interactions of *tif-1A* RNAi with *ife-2* RNAi and *tif-1A* RNAi with CHX. It is possible that there is no further lifespan increase, because the *ife-2* RNAi and CHX translation reduction is much smaller than the one caused by *tif-1A/rpoa-2* RNAi and becomes dominated by

the latter. Ideally one should provide evidence similar to Figure S6.a to clarify and assess the extent of translation inhibition also in *ife-2* RNAi, CHX and respective interactions. However, this manuscript already went through multiple revisions. In order to not further delay the publication, one may adopt the wording accordingly and discuss the limitations in the Discussion section.

Response: We thank the reviewer for their positive assessment of our study. We agree with the reviewer that the respective contributions of translation and metabolism to the longevity effects of the reduced Pol I activity are hard to completely disentangle. Specifically, we do not have data to directly compare the degree of translational repression in *ife-2* mutants, CHX treated animals and *tif-1A/rpoa-2* RNAi exposed worms. We therefore amended the text (page 10, lines 4-7 and page 20, lines 5-10) by including the acknowledgement of this limitation.

On page 10 line 3 "indicating that direct translational repression does not preserve cellular energy as efficiently as inhibition of pre-rRNA synthesis, which in turn affects all subsequent steps of ribosomal biogenesis". For that we would need to know if translation is being equally repressed.

Response: Similar to the above response, we agree with the reviewer, and added amendments to the respective sections of the manuscript (page 10, lines 13-18 and page 20, lines 5-10).

2. rRNA inhibition sustains mitochondrial function?

The authors conclude from the metformin experiments (page 10/11): "This result indicated that restriction of rRNA synthesis sustains mitochondrial function and metabolic plasticity during aging in *C. elegans*."

I don't think the specific metformin experiments show that mitochondrial function is better maintained. I think they only confirm the increased metabolic plasticity. I suggest to word the sentence as follows:

"This result indicated that restriction of rRNA synthesis sustains energy metabolism and metabolic plasticity during aging in *C. elegans*."

Response: We agree, and the respective sections of the manuscript were amended to accommodate the reviewer's request (page 11, lines 13-15 and page 20, lines 16-19).

Minor points

3. Figure 5a: label on the x-axis is missing.

Response: The labelling of the axis was updated.

4. Figure S4/S5 has not only an unnecessary but also meaningless x-axis if you just call it "bubbles_z" without further explanation.

Response: The automatically added Z-score labelling is indeed not needed in these figures, and it was removed. We thank the reviewer for pointing this out.

Reviewer #6 (Remarks to the Author):

The authors addressed the majority of the comments of this Reviewer in the previously revised manuscript. However, the Reviewer is still concerned about the fly *Drosophila* experiments and indicated some important controls (via standardized fly backcrossing) were missing to substantiate the lifespan fly data. The Reviewer states in the current form the fly data are therefore not solid enough for publication and suggested to remove them since they are not essential to the manuscript. The authors indicated there has been a misunderstanding on the original Reviewers request and agreed to remove the fly data from the manuscript.

Response: We thank the reviewer for their favourable assessment and for acknowledging our compliance with the reviewer's requests.